# The 3-phosphoinositide–dependent protein kinase 1 is an essential upstream activator of protein kinase A in malaria parasites

Eva Hitz[1,2], Natalie Wiedemar[1,2¤a], Armin Passecker[1,2], Beatriz A. S. Graça[1,2], Christian Scheurer[1,2], Sergio Wittlin[1,2], Nicolas M. B. Brancucci[1,2], Ioannis Vakonakis[3¤b], Pascal Mäser[1,2], Till S. Voss[1,2]*

1 Department of Medical Parasitology and Infection Biology, Swiss Tropical and Public Health Institute, Basel, Switzerland, 2 University of Basel, Basel, Switzerland, 3 Department of Biochemistry, University of Oxford, Oxford, United Kingdom

¤a Current address: School of Life Sciences, University of Dundee, Dundee, United Kingdom
¤b Current address: Lonza Biologics, Lonza Ltd, Visp, Switzerland
* till.voss@swisstph.ch

**Data Availability Statement:** All relevant data are within the paper and its Supporting Information files. Whole genome sequencing data was

## Abstract

Cyclic adenosine monophosphate (cAMP)-dependent protein kinase A (PKA) signalling is essential for the proliferation of *Plasmodium falciparum* malaria blood stage parasites. The mechanisms regulating the activity of the catalytic subunit PfPKAc, however, are only partially understood, and PfPKAc function has not been investigated in gametocytes, the sexual blood stage forms that are essential for malaria transmission. By studying a conditional PfPKAc knockdown (cKD) mutant, we confirm the essential role for PfPKAc in erythrocyte invasion by merozoites and show that PfPKAc is involved in regulating gametocyte deformability. We furthermore demonstrate that overexpression of PfPKAc is lethal and kills parasites at the early phase of schizogony. Strikingly, whole genome sequencing (WGS) of parasite mutants selected to tolerate increased PfPKAc expression levels identified missense mutations exclusively in the gene encoding the parasite orthologue of 3-phosphoinositide–dependent protein kinase-1 (PfPDK1). Using targeted mutagenesis, we demonstrate that PfPDK1 is required to activate PfPKAc and that T189 in the PfPKAc activation loop is the crucial target residue in this process. In summary, our results corroborate the importance of tight regulation of PfPKA signalling for parasite survival and imply that PfPDK1 acts as a crucial upstream regulator in this pathway and potential new drug target.

## Introduction

Malaria is caused by protozoan parasites of the genus *Plasmodium*. Infections with *Plasmodium falciparum* are responsible for the vast majority of severe and fatal malaria cases. People get infected through female *Anopheles* mosquitoes that inject sporozoites into the skin tissue during their blood meal. After reaching the liver, sporozoites infect and multiply inside hepatocytes, generating thousands of merozoites that are released into the blood stream. Merozoites

deposited on the European Nucleotide Archive, accession number PRJEB40033.

**Funding:** This work was supported by funding from the Swiss National Science Foundation (https://www.snf.ch/en) to TSV (grant number BSCGI0_157729) and PM (grant number 310030_156264), and from the Rudolf Geigy Foundation (https://en.geigystiftung.ch/) to EH. BASG received a PhD fellowship from the European Union's Horizon 2020 research and innovation programme under the Marie Skłodowska-Curie grant agreement number 860875 (https://ec.europa.eu/programmes/horizon2020/en/home). IV received funding from the Medical Research Council UK (https://mrc.ukri.org/) (grant number MR/N009274/1) and the EPA Cephalosporin Fund (https://register-of-charities.charitycommission.gov.uk/charity-details/?regid=309698&subid=0) (grant number CF 329). The funders had no role in study design, data collection and analysis, decision to publish, or preparation of the manuscript.

**Competing interests:** The authors have declared that no competing interests exist.

**Abbreviations:** AC, adenylyl cyclase; AMA1, apical membrane antigen 1; BSD, blasticidin deaminase; cAMP, cyclic adenosine monophosphate; CC, choline chloride; cKD, conditional knockdown; cOE, conditional overexpression; ExR, exflagellation rate; GATK, Genome Analysis Toolkit; gDNA, genomic DNA; GlcN, glucosamine; GlcNac, N-acetyl-D-glucosamine; GMQE, Global Model Quality Estimation; hpi, hours postinvasion; HR, homology region; IDC, intraerythrocytic developmental cycle; iRBC, infected red blood cell; KO, knockout; MAPK, mitogen-activated protein kinase; NPP, new permeability pathway; PCM, parasite culture medium; PDE, phosphodiesterase; PDK1, 3-phosphoinositide-dependent protein kinase-1; PH, pleckstrin homology; PI3K, phosphoinositide 3-kinase; PIF, PDK1-interacting fragment; PKA, protein kinase A; RBC, red blood cell; RNA-seq, RNA sequencing; SCR, sexual commitment rate; sgRNA, single guide RNA; SLI, selection-linked integration; uRBC, uninfected red blood cell; WGS, whole genome sequencing; WT, wild-type; XA, xanthurenic acid.

invade red blood cells (RBCs) and develop intracellularly through the ring stage into a trophozoite and finally a schizont stage parasite, which undergoes 4 to 5 rounds of nuclear division followed by cytokinesis to produce up to 32 new merozoites. Upon rupture of the infected red blood cell (iRBC), the released merozoites infect new erythrocytes to initiate another intraerythrocytic developmental cycle (IDC). The consecutive rounds of RBC invasion and intraerythrocytic parasite proliferation are responsible for all malaria-related pathology and deaths. Importantly, during each round of replication, a small subset of trophozoites commits to sexual development, and their ring stage progeny differentiates over the course of 10 to 12 days and 5 distinct morphological stages (stages I to V) into mature male or female gametocytes. Sexual commitment occurs in response to environmental triggers that activate expression of the transcription factor PfAP2-G, the master regulator of sexual conversion [1–3]. When taken up by an *Anopheles* mosquito, mature stage V gametocytes develop into gametes and undergo fertilisation. The resulting zygote develops into an ookinete that migrates through the midgut wall and transforms into an oocyst, generating thousands of infectious sporozoites ready to be injected into another human host.

Erythrocyte invasion by merozoites is a highly regulated multistep process starting with the initial attachment of the merozoite to the RBC surface, followed by parasite reorientation and formation of a so-called tight junction [4]. The tight junction is the intimate contact area between the merozoite and RBC membranes that moves along the merozoite surface during the actin–myosin motor-driven invasion process [4]. Alongside the secreted rhoptry neck proteins, the micronemal transmembrane protein apical membrane antigen 1 (AMA1) is an integral component of the tight junction [4]. The cytoplasmic domain of AMA1 bears an essential role in merozoites during RBC invasion [5–7]. In particular, the phosphorylation of residues in the AMA1 cytoplasmic tail (S610 and T613) is essential for AMA1 function in RBC invasion [5–7]. Recent research has shown that the *P. falciparum* cyclic adenosine monophosphate (cAMP)-dependent protein kinase A (PfPKA) is responsible for AMA1 phosphorylation at S610 and hence essential for successful erythrocyte invasion [5,6,8,9].

Protein kinase A (PKA) was discovered in the 1970s and is one of the best characterised eukaryotic protein kinases [10]. In its inactive state, the PKA holoenzyme is a tetramer consisting of 2 regulatory subunits (PKAr) and 2 catalytic subunits (PKAc) [11]. Upon binding of cAMP to PKAr, the PKAc subunits are released. PKAc release thus depends on cAMP levels, which are regulated by adenylyl cyclases (ACs) and phosphodiesterases (PDEs) that synthesise and hydrolyse cAMP, respectively [11]. Furthermore, phosphorylation of PKAc is essential for its activity. In vitro, PKAc was shown to be active upon release from PKAr due to autophosphorylation [12,13]. However, research in budding yeast and human cells demonstrated that the 3-phosphoinositide–dependent protein kinase-1 (PDK1) phosphorylates and activates PKAc in vivo [14–18]. PDK1 has originally been identified as the kinase responsible for activating PKB/Akt in response to growth factor–induced phosphoinositide 3-kinase (PI3K) signalling in human cells [17,19–21]. Subsequent studies have shown that PDK1 also activates a large number of other AGC type kinases including PKA, PKG, and PKC [17,21]. AGC kinases dock with PDK1 via their so-called PDK1-interacting fragment (PIF), a hydrophobic motif that binds to the PIF-binding pocket in the N-terminal region of the PDK1 kinase domain, and this interaction allows PDK1 to activate its substrates by activation loop phosphorylation [17,21,22]. In case of human PKAc, the PDK1-dependent phosphorylation of T197 in the activation loop plays a crucial role in controlling PKAc structure, activity, and function [15,18,23,24].

In *P. falciparum*, PfPKA kinase consists of only one catalytic (PfPKAc) and one regulatory (PfPKAr) subunit [25,26], and cAMP levels in blood stage parasites are regulated by PfPDEβ [27], which hydrolyses both cAMP and cGMP and PfACβ that synthesises cAMP [8]. Analysis

of conditional loss-of-function mutants showed that PfPKAc is essential for the process of merozoite invasion, where it is required for the phosphorylation and timely shedding of the invasion ligand AMA1 from the merozoite surface [5,6,8,9]. Likewise, depletion of cAMP levels through conditional disruption of *pfacβ* phenocopied the invasion defect observed for the *pfpkac* mutant [8]. Interestingly, a conditional *pfpdeβ* null mutant, which displays increased cAMP levels and PfPKAc hyperactivation, also showed a severe merozoite invasion defect that was linked to elevated phosphorylation and premature shedding of AMA1 [27]. These studies highlighted that tight regulation of PfPKAc activity is crucial for successful merozoite invasion and parasite proliferation. In addition, PfPKA seems to have additional functions in blood stage parasites. Global phosphoproteomic studies of *pfpdeβ*, *pfacβ*, and *pfpkac* conditional knockout (KO) cell lines identified 39 proteins as high confidence targets of cAMP/PfPKA-dependent phosphorylation [8,27]. These proteins include invasion factors (e.g., AMA1 and coronin) and several proteins with predicted roles in other processes (e.g., chromatin organisation and protein transport) or with unknown functions [8,27]. In addition, cAMP/PfPKA-dependent signalling has been implicated in the regulation of ion channel conductance and new permeability pathways (NPPs) in asexual blood stage parasites [28] and gametocytes [29] as shown through the use of pharmacological approaches (PKA/PDE inhibitors, exogenous 8-Bromo-cAMP) and transgenic cell lines (deletion of PfPDEδ, overexpression of PfPKAr) [28,29]. Similar experiments identified a putative role for cAMP/PfPKA-dependent signalling in regulating gametocyte-infected erythrocyte deformability [30].

While several studies demonstrated the importance of cAMP in activating PfPKAc, the role of PfPKAc phosphorylation in regulating PfPKAc activity remains elusive. High-throughput phosphoproteomic approaches identified several phosphorylated residues in PfPKAc, including T189 that corresponds to the PDK1 target residue T197 in the activation segment of mammalian PKAc [31–34]. However, if and to what extent phosphorylation of T189 is important for PfPKAc activation in *P. falciparum* and whether T189 phosphorylation is deposited via autophosphorylation or by another kinase, is unknown. Furthermore, besides the well-established role for PfPKAc in parasite invasion, other possible functions of PfPKA in asexual and sexual development are only poorly understood.

Here, we used reverse genetics approaches to study the function of PfPKAc in asexual blood stage parasites, sexual commitment, and gametocytogenesis. Our results confirm the essential role for PfPKAc in merozoite invasion and show that while PfPKAc plays no obvious role in the control of sexual commitment or gametocyte maturation, it contributes to the regulation of gametocyte-infected erythrocyte deformability. We further demonstrate that overexpression of PfPKAc is lethal in asexual blood stage parasites. Intriguingly, whole genome sequencing (WGS) of parasites selected to tolerate PfPKAc overexpression identified mutations exclusively in the gene encoding the *P. falciparum* orthologue of phosphoinositide-dependent protein kinase-1 (PfPDK1). Using targeted mutagenesis, we show that the T189 residue is crucial for PfPKAc activity and that activation of PfPKAc is likely dependent on PfPDK1-mediated phosphorylation.

## Results

### Generation of a conditional PfPKAc loss-of-function mutant

To study PfPKAc function, we generated a conditional PfPKAc knockdown (cKD) line using a selection-linked integration (SLI)-based gene editing approach [35]. Successful engineering tags the *pfpkac* gene with the fluorescent marker gene *gfp* fused to the *fkbp* destabilisation domain (*dd*) sequence [36,37], followed by a sequence encoding the 2A skip peptide [38,39], the blasticidin S deaminase (*bsd*) marker gene, and, finally, the *glmS* ribozyme element [40]

(S1 Fig). The FKBP/DD system allows for protein destabilisation in the absence of the small molecule ligand Shield-1 [36,37], whereas the *glmS* ribozyme in the 3′ untranslated region of the mRNA causes transcript degradation in the presence of glucosamine (GlcN) [40]. To be able to easily quantify sexual commitment rates (SCRs), we modified the *pfpkac* gene in NF54/ AP2-G-mScarlet parasites [41], which allows identifying sexually committed cells by visualising expression of fluorophore-tagged PfAP2-G [3,42]. After drug selection of transgenic NF54/AP2-G-mScarlet/PKAc cKD parasites, correct editing of *pfpkac* and absence of the wild-type (WT) locus was confirmed by PCR on genomic DNA (gDNA) (S1 Fig). Live cell fluorescence imaging demonstrated that PfPKAc-GFPDD was expressed in the cytosol and nucleus in mid schizonts before it localised to the periphery of developing merozoites in late schizonts as also described elsewhere [8,9] (S1 Fig). PfPKAc-GFPDD expression was efficiently depleted in parasites cultured under–Shield-1/+GlcN conditions compared to the matching control (+-Shield-1/−GlcN) (Fig 1A). To assess the growth phenotype of PfPKAc-GFPDD-depleted parasites, we split ring stage parasite cultures (0 to 6 hours postinvasion, hpi), maintained them separately under PfPKAc-GFPDD-depleting (−Shield-1/+GlcN) and PfPKAc-GFPDD-stabilising conditions (+Shield-1/−GlcN) and quantified parasitaemia over 3 generations using flow cytometry (Figs 1B and S2). As expected, PfPKAc-GFPDD-depleted parasites were unable to proliferate because the merozoites failed to invade new RBCs (Fig 1C). Hence, by combining 2 inducible expression systems, we engineered a PfPKAc cKD mutant that allows efficient depletion of PfPKAc-GFPDD expression and phenocopies the lethal invasion defect previously observed with conditional PfPKAc KO mutants [89].

## PfPKAc plays no major role in sexual commitment and gametocyte maturation but contributes to the regulation of gametocyte rigidity

To investigate whether PfPKAc activity regulates sexual commitment, we split ring stage parasites (0 to 6 hpi) and cultured them separately under PfPKAc-GFPDD-depleting (−Shield-1/ +GlcN) and PfPKAc-GFPDD-stabilising conditions (+Shield-1/−GlcN). Sexually committed parasites were identified based on PfAP2-G-mScarlet positivity in late schizonts (40 to 46 hpi) using high content imaging. We observed a significant increase in SCRs in PfPKAc-GFPDD-depleted (−Shield-1/+GlcN) compared to control parasites (+Shield-1/−GlcN) (1.56-fold ± 0.13 SD) (S3 Fig). We can exclude that these differences are caused by the Shield-1 compound as we have previously shown that Shield-1 treatment has no effect on SCRs [43]. However, treatment with GlcN also caused a similar increase in SCRs in NF54/AP2-G-mScarlet control parasites (1.44-fold ± 0.14 SD) (S3 Fig). We therefore conclude that PfPKAc plays no major role in regulating sexual commitment and that the increased SCRs observed in PfPKAc-GFPDD-depleted parasites are caused by the presence of 2.5 mM GlcN in the culture medium.

Next, we tested if PfPKAc-GFPDD depletion affects gametocyte morphology or male gametocyte exflagellation. To conduct these experiments, we induced sexual commitment by culturing parasites in serum-free minimal fatty acid medium (−SerM) [3]. After reinvasion, parasites were split and cultured separately under PfPKAc-GFPDD-depleting (−Shield-1/+GlcN) or PfPKAc-GFPDD-stabilising conditions (+Shield-1/−GlcN). For the first 6 days of gametocyte maturation, the growth medium was supplemented with 50 mM N-acetyl-D-glucosamine (GlcNac) to eliminate asexual parasites [44]. Despite efficient depletion of PfPKAc-GFPDD expression (Figs 1D and S4), we could not detect morphological abnormalities in PfPKAc-GFPDD-depleted gametocytes in any of the 5 developmental stages (I to V) based on visual inspection of Giemsa-stained thin blood smears (S4 Fig). While the exflagellation rates (ExRs) of PfPKAc-GFPDD-depleted male stage V gametocytes (−Shield-1/+GlcN) were significantly

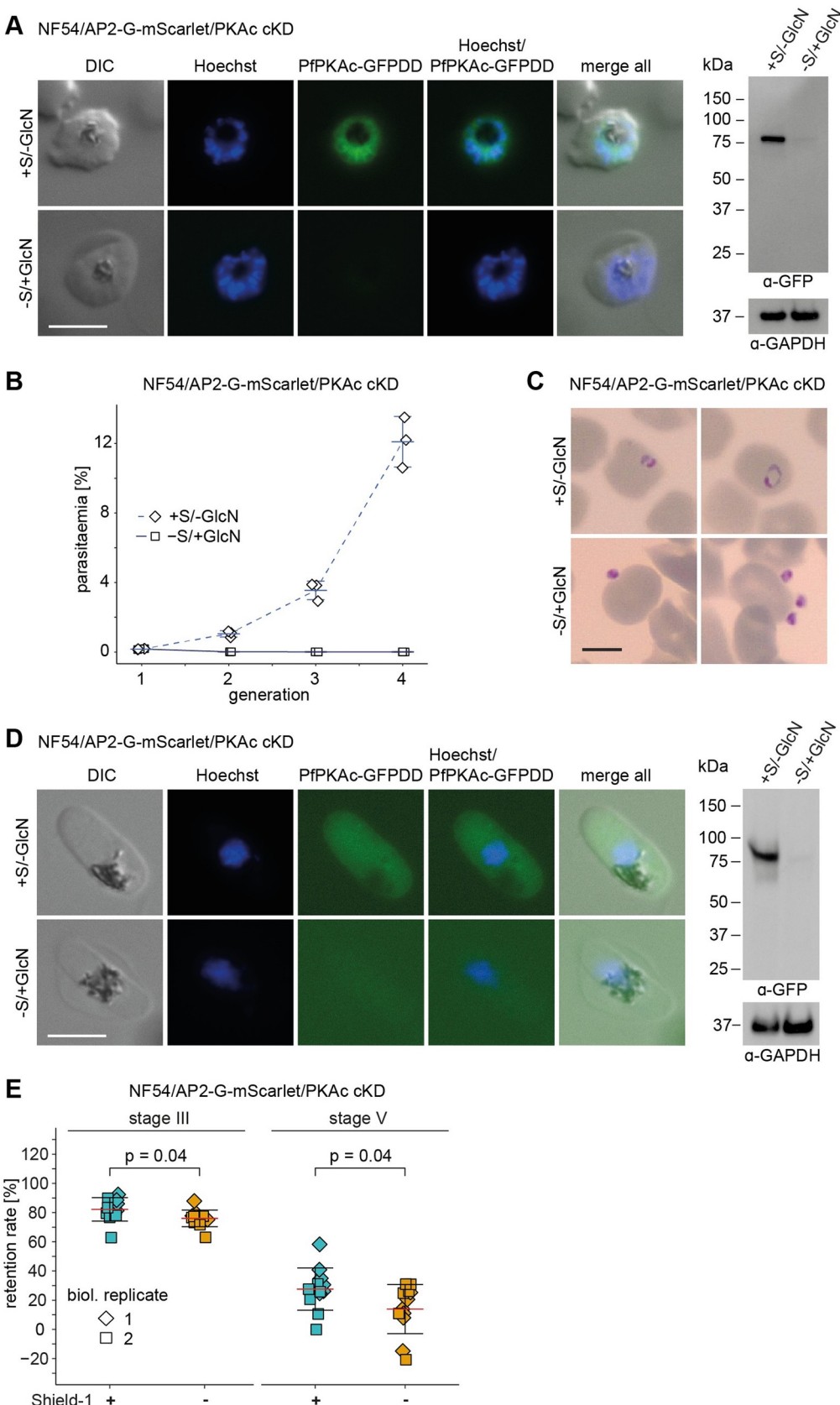

**Fig 1. Depletion of PfPKAc in NF54/AP2-G-mScarlet/PKAc cKD parasites leads to a block in merozoite invasion and decreases gametocyte rigidity. (A)** Expression of PfPKAc-GFPDD in late schizonts under protein- and RNA-depleting (−Shield-1/+GlcN) and control conditions (+Shield-1/−GlcN) as assessed by live cell fluorescence imaging and western blot analysis. Synchronous parasites (0 to 8 hpi) were split (±Shield-1/±GlcN) 40 hours before sample collection. Representative fluorescence images are shown. Parasite DNA was stained with Hoechst. Scale bar = 5 μm. For western blot analysis, parasite lysates derived from equal numbers of parasites were loaded per lane. MW PfPKAc-GFPDD = 79.8 kDa, MW PfGAPDH = 36.6 kDa. The full size western blot is shown in S1 Fig. **(B)** Increase in parasitaemia over 3 generations under PfPKAc-GFPDD-depleting (−Shield-1/+GlcN) and control conditions (+Shield-1/−GlcN). Open squares represent data points for individual replicates and the means and SD (error bars) of 3 biological replicates are shown. The raw data are available in the source data file (S2 Data). **(C)** Representative images from Giemsa-stained thin blood smears showing the progeny of parasites cultured under PfPKAc-GFPDD-depleting (−Shield-1/+GlcN) or control conditions (+Shield-1/−GlcN) conditions at 0 to 6 hpi (generation 2). Scale bar = 5 μm. **(D)** Expression of PfPKAc-GFPDD in stage V gametocytes (day 11) under protein- and RNA-depleting (−Shield-1/+GlcN) and control conditions (+Shield-1/−GlcN) as assessed by live cell fluorescence imaging and western blot analysis. Representative fluorescence images are shown. Parasite DNA was stained with Hoechst. Scale bar = 5 μm. For western blot analysis, parasite lysates derived from equal numbers of parasites were loaded per lane. MW PfPKAc-GFPDD = 79.8 kDa, MW PfGAPDH = 36.6 kDa. The full size western blot is shown in S4 Fig. **(E)** Retention rates of stage III (day 6) and stage V (day 11) gametocytes cultured under PfPKAc-GFPDD-depleting (−Shield-1) (orange) and control conditions (+Shield-1) (blue). Coloured squares represent data points for individual replicates and the means and SD (error bars) of 2 biological replicates performed in 6 technical replicates each are shown. Differences in retention rates have been compared using an unpaired 2-tailed Student $t$ test (statistical significance cutoff: $p < 0.05$). The raw data are available in the source data file (S2 Data). cKD, conditional PfPKAc knockdown; DIC, differential interference contrast; GlcN, glucosamine; hpi, hours postinvasion; PKA, protein kinase A; S, Shield-1.

reduced by more than 50% compared to the control (+Shield-1/−GlcN), this effect was again caused by the presence of GlcN in the culture medium (S4 Fig). When PfPKAc-GFPDD expression was depleted by Shield-1 removal only (−Shield-1/−GlcN), no significant differences in ExRs were observed even though PfPKAc-GFPDD expression was efficiently reduced (S4 Fig). In line with this result, GlcN treatment also led to a significant reduction in ExRs of NF54 WT gametocytes (+GlcN) to less than 50% compared to the control (−GlcN) (S4 Fig).

Lastly, we tested if PfPKAc activity is involved in regulating gametocyte rigidity. Immature gametocytes display high cellular rigidity and sequester in the bone marrow and the spleen, whereas stage V gametocytes are more deformable and can reenter the bloodstream to be taken up by feeding mosquitoes [45–47]. Experiments employing PKA and PDE inhibitors, a transgenic cell line overexpressing PfPKAr, or treatment with exogenous 8-bromo-cAMP to increase cellular cAMP levels provided evidence for a potential role for PfPKAc in maintaining the rigidity of immature gametocyte-infected erythrocytes [30]. To test if PfPKAc is indeed involved in controlling this process, we measured the deformability status of immature stage III and mature stage V gametocytes using microsphiltration experiments [46]. Microsphiltration exploits the fact that differences in cellular rigidity correlate with cell retention rates in a microsphere-based artificial spleen system [46,47]. We observed a slight but significant decrease in the retention rates of PfPKAc-GFPDD-depleted gametocytes (−Shield-1) compared to the control (+Shield-1), both in immature stage III (day 6) (76.0% ± 5.7 SD versus 82.2% ± 8.0 SD) and mature stage V (day 11) gametocytes (13.9% ± 16.8 SD versus 27.6% ± 14.5 SD) (Fig 1E). By contrast, NF54 WT gametocytes cultured in the presence or absence of Shield-1 showed no difference in retention rates (S4 Fig).

In summary, our results demonstrate that PfPKAc plays no major role in the regulation of sexual commitment, gametocyte maturation, or male gametogenesis but that gametocyte-iRBC rigidity is at least partially regulated by PfPKAc. Furthermore, we discovered that the presence of 2.5 mM GlcN in the culture medium affects both sexual commitment and ExRs, which needs to be taken into account when studying these processes in conditional mutants employing the *glmS* riboswitch system.

## PfPKAc overexpression in asexual blood stage parasites is lethal

The merozoite invasion and post-invasion developmental defects observed for *pfpdeβ* KO parasites had been linked to increased cAMP levels and PfPKAc hyperactivity [27]. To further study the consequences of PfPKAc overexpression, we generated a PfPKAc conditional overexpression (cOE) line using CRISPR/Cas-9–based gene editing. To this end, we inserted an ectopic *pfpkac-gfp* transgene cassette into the nonessential *glp3* (*cg6*, Pf3D7_0709200) locus in NF54 WT parasites (NF54/PKAc cOE) (S5 Fig). Here, the constitutive *calmodulin* (PF3D7_1434200) promoter and a *glmS* ribozyme element [48] control expression of the *pfpkac-gfp* gene. Since the initial transgenic NF54/PKAc cOE population still contained some parasites carrying the WT *glp3* locus, we obtained clonal lines by limiting dilution cloning [49]. In 2 clones (NF54/PKAc cOE M1 and M2), correct integration of the inducible PfPKAc-GFP OE expression cassette and absence of WT parasites was confirmed by PCR on gDNA (S5 Fig). Live cell fluorescence imaging and western blot analysis confirmed the efficient induction of PfPKAc-GFP OE upon GlcN removal in both clones (Figs 2A and S6). Interestingly, PfPKAc-GFP OE (–GlcN) resulted in a complete block in parasite development half way through the IDC (Fig 2B). To study this growth defect in more detail, we quantified the number of nuclei per parasite in late schizont stages (40 to 46 hpi), 40 hours after triggering PfPKAc-GFP OE (–GlcN) in young ring stage parasites (0 to 6 hpi). This experiment revealed that NF54/PKAc cOE M1 parasites overexpressing PfPKAc-GFP did not develop beyond the late trophozoite/early schizont stage as most parasites contained only 1 or 2 nuclei as opposed to the control population (+GlcN) that progressed normally through several rounds of nuclear division (Fig 2C). To test whether these parasites were still able to produce progeny, we performed parasite multiplication assays. NF54/PKAc cOE M1 ring stage parasites were split at 0 to 6 hpi, cultured separately in the presence or absence of GlcN and parasite multiplication was quantified over 3 generations. PfPKAc-GFP OE (–GlcN) completely failed to multiply as no increase in parasitaemia was observed, in contrast to the control population (+GlcN) that multiplied normally (Figs 2D and S6). Notably, however, viable PfPKAc-GFP OE parasites emerged approximately 2 weeks after maintaining NF54/PKAc cOE M1 parasites constantly in culture medium lacking GlcN. We termed these parasites "PfPKAc OE survivors" (NF54/PKAc cOE S1). Importantly, NF54/PKAc cOE S1 parasites still overexpressed PfPKAc-GFP in absence of GlcN (Figs 2E and S6). Furthermore, Sanger sequencing confirmed that neither the endogenous nor the ectopic *pfpkac* genes in the NF54/PKAc cOE S1 survivor population carried any mutations. Quantifying the number of nuclei per schizont and parasite multiplication assays revealed that NF54/PKAc cOE S1 parasites were completely tolerant to PfPKAc-GFP OE and developed and multiplied identically irrespective of whether PfPKAc-GFP OE was induced (–GlcN) or not (+GlcN) (Figs 2F, 2G and S6).

In conclusion, PfPKAc OE causes a lethal phenotype in asexual parasites by preventing parasite development beyond the late trophozoite/early schizont stage. However, parasites tolerant to PfPKAc-GFP OE can be selected for, and these parasites show no defect in intraerythrocytic development and parasite multiplication.

## Parasites tolerant to PfPKAc overexpression carry mutations in the gene encoding *P. falciparum* 3-phosphoinositide–dependent protein kinase-1 (PfPDK1)

The above findings suggested that genetic mutations in the NF54/PKAc cOE S1 survivor population might cause their tolerance to elevated PfPKAc-GFP expression levels. To address this hypothesis, we performed WGS of the 2 parental NF54/PKAc cOE clones M1 and M2 and the 6 independently grown survivor populations (NF54/PKAc cOE S1-S6), 3 each originating

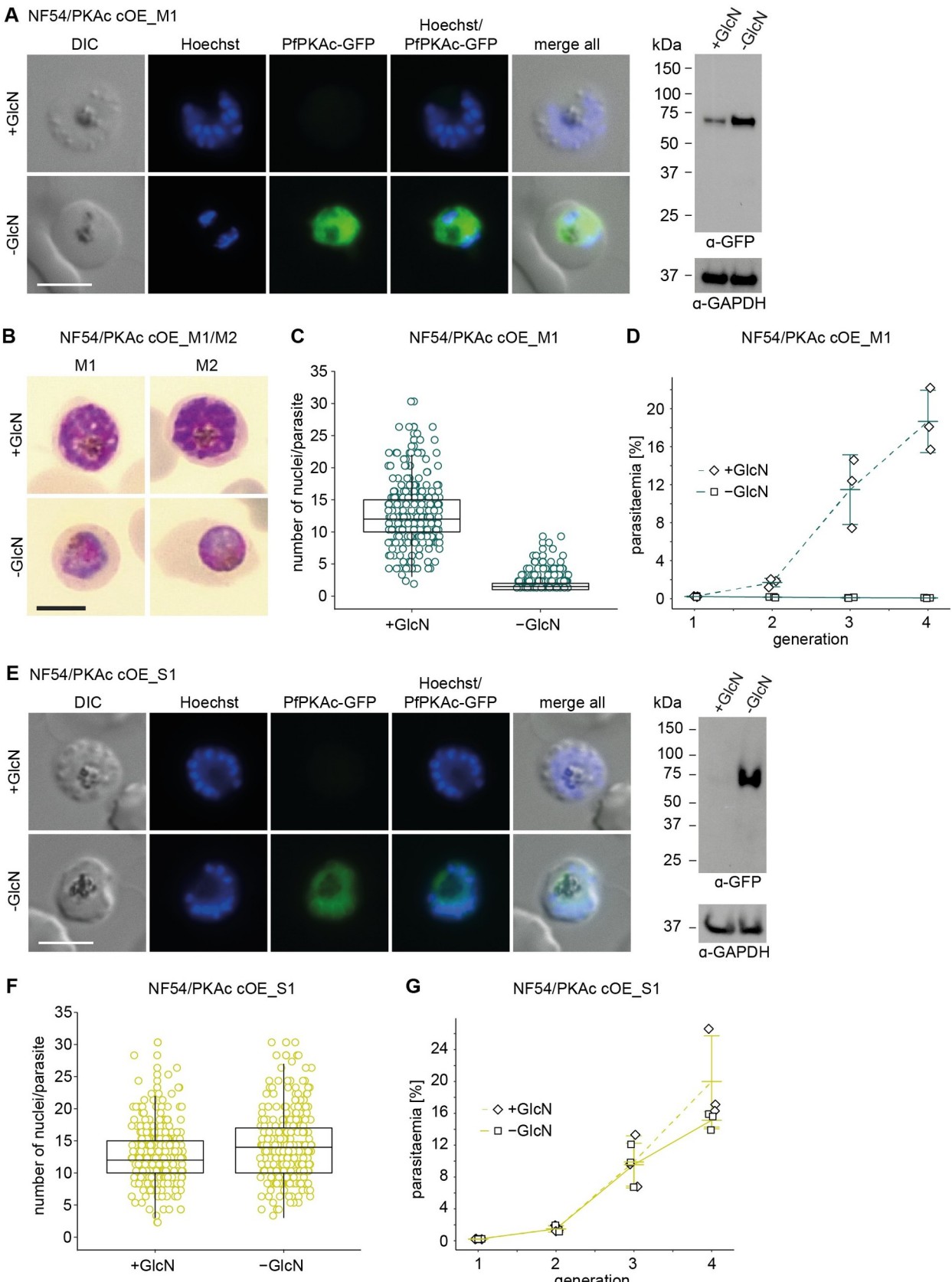

**Fig 2. Overexpression of PfPKAc in NF54/PKAc cOE M1 parasites is lethal but survivor populations tolerant to PfPKAc OE can be selected.** **(A)** Expression of PfPKAc-GFP in NF54/PKAc cOE M1 parasites under OE-inducing (−GlcN) and control conditions (+GlcN) as assessed by live cell fluorescence imaging and western blot analysis. Synchronous parasites (0 to 8 hpi) were split (±GlcN) 40 hours before sample collection. Representative fluorescence images are shown. Parasite DNA was stained with Hoechst. Scale bar = 5 μm. For western blot analysis, lysates derived from equal numbers of parasites were loaded per lane. MW PfPKAc-GFP = 67.3 kDa, MW PfGAPDH = 36.6 kDa. The full size western blot is shown in S6 Fig. **(B)** Representative images from Giemsa-stained thin blood smears showing NF54/PKAc cOE M1 and M2 parasites under PfPKAc-GFP OE-inducing (−GlcN) and control conditions (+GlcN). Synchronous parasites (0 to 8 hpi) were split (±GlcN) 40 hours before the images were captured. Scale bar = 5 μm. **(C)** Number of nuclei per schizont in NF54/PKAc cOE M1 parasites under PfPKAc-GFP OE-inducing (−GlcN) and control conditions (+GlcN). Each open circle represents one parasite. Data from 3 biological replicate experiments are shown, and 100 parasites were counted in each experiment. The boxplots show data distribution (median, upper, and lower quartile and whiskers). The raw data are available in the source data file (S2 Data). **(D)** Increase in parasitaemia in NF54/PKAc cOE M1 parasites over 3 generations under PfPKAc-GFP OE-inducing (−GlcN) and control (+GlcN) conditions. Synchronous parasites (0 to 6 hpi) were split (±GlcN) 18 hours before the first measurement in generation 1. Open squares represent data points for individual replicates and the means and SD (error bars) of 3 biological replicates are shown. The raw data are available in the source data file (S2 Data). **(E)** Expression of PfPKAc-GFP in NF54/PKAc cOE S1 survivor parasites under OE-inducing (−GlcN) and control conditions (+GlcN) as assessed by live cell fluorescence imaging and western blot analysis. Parasites were cultured and samples prepared as described in panel A. Scale bar = 5 μm. MW PfPKAc-GFP = 67.3 kDa, MW PfGAPDH = 36.6 kDa. The full size western blot is shown in S6 Fig. **(F)** Number of nuclei per schizont in NF54/PKAc cOE S1 survivor parasites under PfPKAc-GFP OE-inducing (−GlcN) and control conditions (+GlcN). Each open circle represents one parasite. Data from 3 biological replicate experiments are shown, and 100 parasites were counted in each experiment. The boxplots show data distribution (median, upper, and lower quartile and whiskers). The raw data are available in the source data file (S2 Data). **(G)** Increase in parasitaemia in NF54/PKAc cOE S1 survivor parasites over 3 generations under PfPKAc-GFP OE-inducing (−GlcN) and control (+GlcN) conditions. Parasites were cultured as described in panel D. Open squares represent data points for individual replicates and the means and SD (error bars) of 3 biological replicates are shown. The raw data are available in the source data file (S2 Data). cOE, conditional overexpression; DIC, differential interference contrast; GlcN, glucosamine.

from the M1 and M2 clones, respectively. Intriguingly, we found that all 6 NF54/PKAc cOE survivors, but not the parental M1 and M2 clones, carried missense mutations in the gene encoding a putative serine/threonine protein kinase (Pf3D7_1121900) (Fig 3A, S1 Data). No other mutations were identified in the NF54/PKAc cOE survivor populations, consistent with the Sanger sequencing results, neither the endogenous nor the ectopic *pfpkac* gene carried mutations in any of the 6 survivors. The *Plasmodium vivax* orthologue of Pf3D7_1121900 (PVX_091715) is annotated as putative 3-phosphoinositide dependent protein kinase-1 (PDK1) (www.plasmodb.org), and a multiple sequence alignment suggested that Pf3D7_1121900 is indeed an orthologue of PDK1, a kinase widely conserved among eukaryotes and known as a master regulator of AGC kinases including PKA [17,50] (S7 Fig). However, similar to PDK1 enzymes from most fungi, nonvascular plants, and other alveolates, Pf3D7_1121900 and its *P. vivax* orthologue lack the carboxyl-terminal phospholipid-binding pleckstrin homology (PH) domain that is found in PDK1s from animals and vascular plants and important to localise PDK1 to the plasma membrane for PKB activation in response to the PI3K-dependent production of phosphatidylinositol bis-/trisphosphates [17,50,51]. Construction of a homology model of the Pf3D7_1121900 protein kinase domain based on the human PDK1 crystallographic structure (PDB ID 1UU9) [52] allowed us to visualise the location of the amino acids mutated in the 6 NF54/PKAc cOE survivors (Fig 3B). None of these mutations affected the putative PIF-binding pocket. Rather, all mutated amino acids were located at or in the periphery of the predicted ATP-binding cleft, although only one mutation (N45S) may influence ATP coordination directly (Figs 3B and S7). We hence surmise that all identified mutations alter the catalytic activity but not the protein interaction preferences of PfPDK1.

In conclusion, we identified missense mutations in the Pf3D7_1121900 gene in 6 independently obtained NF54/PKAc cOE survivor parasite lines, which likely confer tolerance to PfPKAc OE. Bioinformatic analysis indicates that this gene encodes PfPDK1, the *P. falciparum* orthologue of PDK1, and modelling of the PfPDK1 structure predicts that the acquired mutations may affect its catalytic activity.

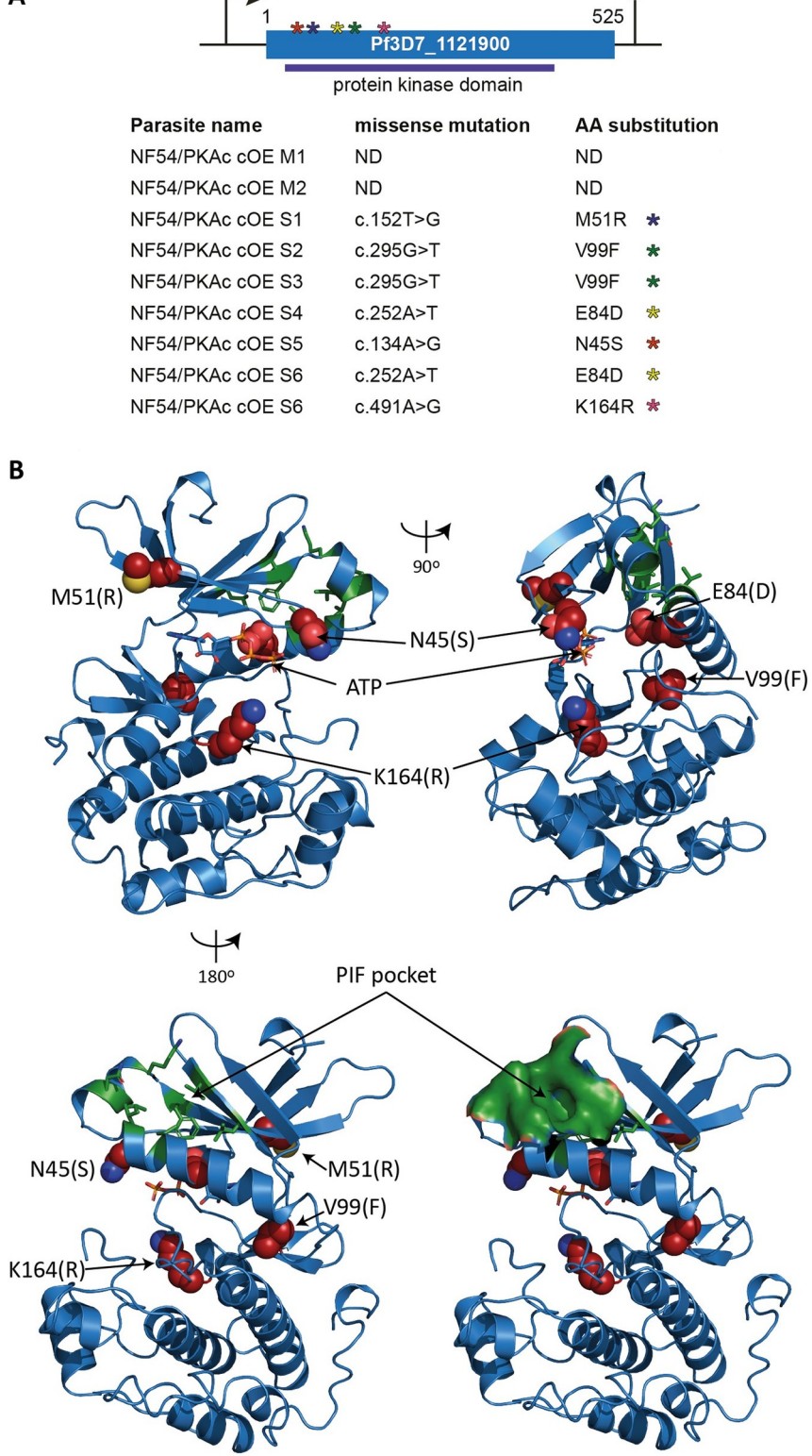

**A**

| Parasite name | missense mutation | AA substitution | |
|---|---|---|---|
| NF54/PKAc cOE M1 | ND | ND | |
| NF54/PKAc cOE M2 | ND | ND | |
| NF54/PKAc cOE S1 | c.152T>G | M51R | * |
| NF54/PKAc cOE S2 | c.295G>T | V99F | * |
| NF54/PKAc cOE S3 | c.295G>T | V99F | * |
| NF54/PKAc cOE S4 | c.252A>T | E84D | * |
| NF54/PKAc cOE S5 | c.134A>G | N45S | * |
| NF54/PKAc cOE S6 | c.252A>T | E84D | * |
| NF54/PKAc cOE S6 | c.491A>G | K164R | * |

**B**

**Fig 3. WGS reveals mutations in the Pf3D7_1121900/*pfpdk1* gene in 6 independently grown NF54/PKAc cOE survivor parasites.** (A) Top: Schematic of the Pf3D7_1121900/*pfpdk1* gene. Asterisks indicate the approximate localisation of the 5 missense mutations identified in the 6 different NF54/PKAc cOE survivors. Bottom: Summary of the sequence information obtained from WGS of gDNA of the 2 NF54/PKAc cOE clones (M1, M2) and the 6

independently grown survivors (S1–S6). Missense mutations and their positions within the *pfpdk1* coding sequence as well as the corresponding amino acid substitutions in the PfPDK1 protein sequence are shown. **(B)** Predicted PfPDK1 structure shown in orthogonal (top left versus top right) or opposing (top left versus lower left) views. PfPDK1 was modelled on the crystallographic structure of human PDK1 (PDB ID 1UU9) [52]. PfPDK1 segments with no correspondence in human PDK1 (amino acids 1 to 27, 188 to 307, and 423 to 525) were omitted from modelling. The ATP substrate (sticks), mutated amino acids (substitution in parenthesis; red spheres), and residues forming the PIF-binding pocket [22] (green sticks) are indicated. The PIF-binding pocket is shown in surface representation in the lower right view. c., cDNA; cOE, conditional overexpression; PIF, PDK1-interacting fragment; WGS, whole genome sequencing.

## Conditional depletion of PfPDK1 has no major effect on parasite multiplication, sexual commitment, and gametocytogenesis

To gain further insight into the function of PfPDK1, we attempted to generate a *pfpdk1* KO line by gene disruption using a CRISPR/Cas-9–based single plasmid approach [53]. However, consistent with the results obtained in previous kinome- or genome-wide KO screens in *P. falciparum* and *Plasmodium berghei* [31,54,55], we failed to obtain a *pfpdk1* null mutant, indicating that the gene is essential in asexual parasites. We therefore engineered the PfPDK1 conditional knockdown line NF54/PDK1 cKD by tagging the endogenous *pfpdk1* gene with g*fp* fused to the *dd* sequence in NF54 WT parasites (S8 Fig). Correct editing of the *pfpdk1* gene and absence of the WT locus was confirmed by PCR on gDNA. Donor plasmid integration downstream of the *pfpdk1* gene was detected in a subset of parasites (S8 Fig), but this is not expected to compromise *pfpdk1-gfpdd* expression since the 551 bp 3′ homology region (HR) used for homology-directed repair includes the native terminator as based on published RNA sequencing (RNA-seq) data [56]. Live cell fluorescence imaging and western blot analysis revealed that PfPDK1-GFPDD is expressed in the cytosol and nucleus throughout the IDC with highest expression in schizonts, consistent with published gene expression data [56,57] (Figs 4A and S9). Furthermore, PfPDK1-GFPDD expression was efficiently reduced after Shield-1 removal compared to parasites grown in the presence of Shield-1 (Figs 4B and S9). Surprisingly, PfPDK1-GFPDD-depleted parasites (−Shield-1) showed slightly higher multiplication rates compared to the control (+Shield-1) (Fig 4C). This increased multiplication rate cannot be attributed to the removal of Shield-1 itself, since NF54 WT parasites cultured in the presence or absence of Shield-1 multiplied equally (S9 Fig). We also tested whether PfPDK1-GFPDD plays a role in gametocytogenesis but did not detect any differences in SCRs, gametocyte morphology or ExRs when comparing PfPDK1-GFPDD-depleted (−Shield-1) with control parasites (+Shield-1) (S10 Fig).

Taken together, we show that PfPDK1 is expressed in the parasite nucleus and cytosol throughout asexual development, reaching peak expression in schizonts. While PfPDK1 is likely essential, the results obtained with the PfPDK1 cKD mutant suggest that residual PfPDK1 expression levels are sufficient to sustain parasite viability. Furthermore, PfPDK1 seems to play no major role in regulating sexual commitment, gametocytogenesis or male gametogenesis, but it is again conceivable that residual PfPDK1 expression in the cKD mutant may have been sufficient to maintain these processes.

## Expression of WT PfPDK1 is incompatible with PfPKAc overexpression, whereas expression of the M51R PfPDK1 mutant causes PfPKAc overexpression tolerance

To test if PfPDK1 is indeed involved in regulating PfPKAc activity, we used CRISPR/Cas-9–based targeted mutagenesis to change the PfPDK1 M51R mutation back to the WT sequence in the NF54/PKAc cOE S1 survivor line (NF54/PKAc cOE S1/PDK1_wt). Sanger sequencing

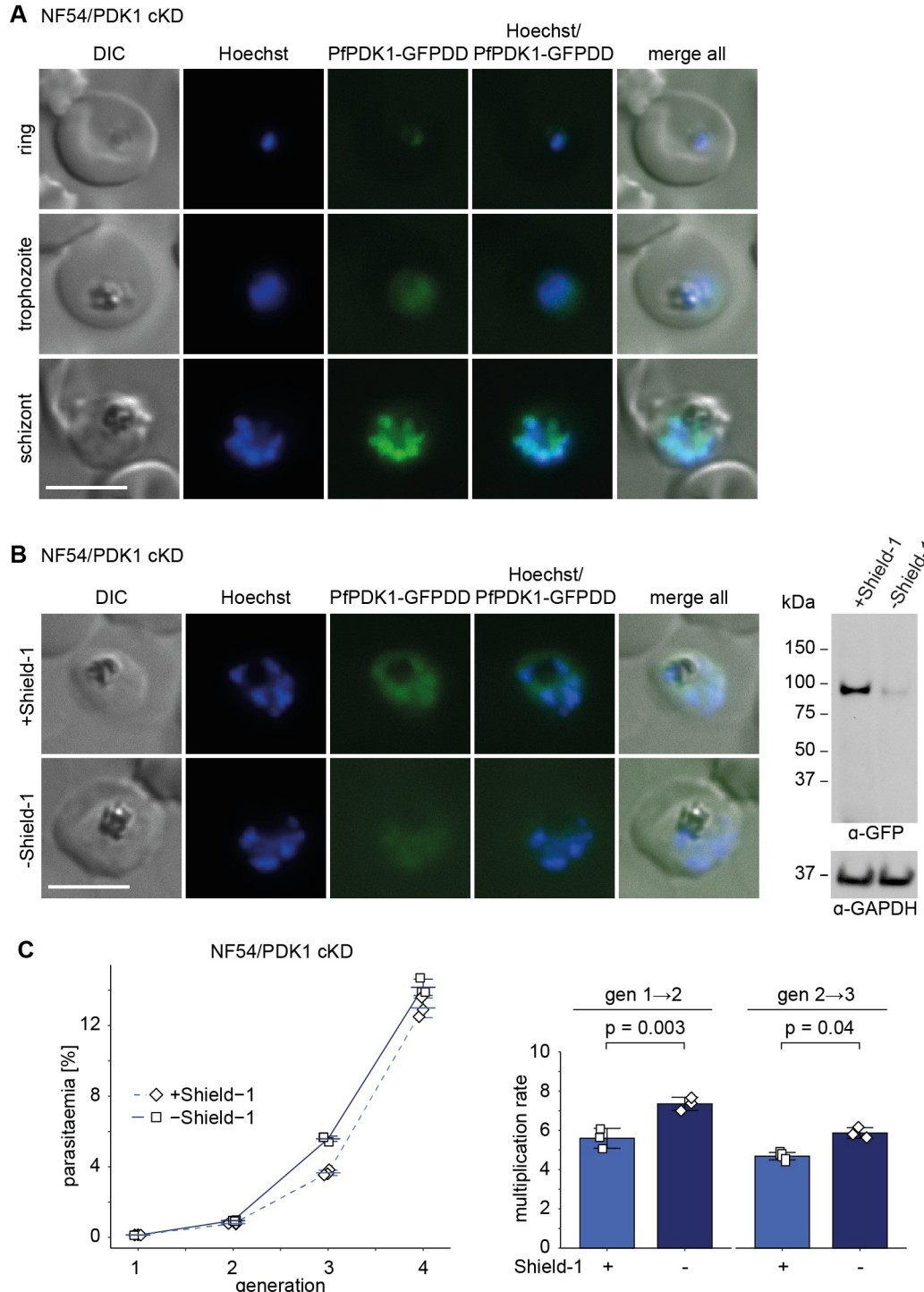

**Fig 4. Knockdown of PfPDK1 expression in NF54/PDK1 cKD parasites has no major effect on asexual parasite growth.** **(A)** Expression of PfPDK1-GFPDD in ring (18 to 24 hpi), trophozoite (24 to 30 hpi), and schizont (42 to 48 hpi) stages cultured under protein-stabilising (+Shield-1) conditions by live cell fluorescence imaging. Representative fluorescence images are shown. Parasite DNA was stained with Hoechst. Scale bar = 5 μm. **(B)** Expression of PfPDK1-GFPDD under protein-depleting (−Shield-1) and control conditions (+Shield-1) by live cell fluorescence imaging and western blot analysis. Synchronous parasites (0 to 8 hpi) were split (±Shield-1) 40 hours before collection of the samples. Representative fluorescence images are shown. Parasite DNA was stained with Hoechst. Scale bar = 5 μm. For western blot analysis, lysates derived from an equal numbers of parasites were loaded per lane. MW PfPDK1-GFPDD = 101.1 kDa, MW PfGAPDH = 36.6 kDa. The full size western blot is shown in S9 Fig. **(C)** Increase in

parasitaemia (left) and parasite multiplication rates (right) under PfPDK1-GFPDD-depleting (–Shield-1) and control (+Shield-1) conditions. Synchronous parasites (0 to 6 hpi) were split (±Shield-1) 18 hours before the first measurement in generation 1. Open squares represent data points for individual replicates and the means and SD (error bars) of 3 biological replicates are shown. Differences in multiplication rates have been compared using a paired 2-tailed Student *t* test (statistical significance cutoff: $p < 0.05$). The raw data are available in the source data file (S2 Data). cKD, conditional PfPKAc knockdown; DIC, differential interference contrast.

verified the successful reversion of the PfPDK1 M51R mutation (S11 Fig), and live cell fluorescence imaging and western blot analysis confirmed that GlcN removal still triggered efficient PfPKAc-GFP OE in NF54/PKAc cOE S1/PDK1_wt parasites (Figs 5A and S11). Strikingly, PfPKAc-GFP OE (–GlcN) led to a complete block in parasite replication, showing that reverting the PfPDK1 M51R mutation completely restored the PfPKAc OE-sensitive phenotype (Figs 5C and S11). To complement these experiments, we also attempted to introduce the PfPDK1 M51R mutation into NF54 WT parasites and the NF54/PKAc cOE clone M1. Two independent attempts to mutate PfPDK1 in NF54 WT parasites failed, suggesting that fully functional PfPDK1 is strictly required for parasite viability under normal PfPKAc expression levels. By contrast, we readily succeeded in introducing the PfPDK1 M51R mutation into the NF54/PKAc cOE clone M1 (NF54/PKAc cOE M1/PDK1_mut) (S11 Fig). Western blot analysis and live cell fluorescence imaging confirmed that GlcN removal still triggered efficient PfPKAc-GFP OE (Figs 5B and S11). Parasite multiplication assays revealed that the PfPDK1 M51R point mutation rendered NF54/PKAc cOE M1/PDK1_mut parasites tolerant to PfPKAc-GFP OE (–GlcN) (Figs 5D and S11). However, in contrast to NF54/PKAc cOE S1 parasites, which carry the same M51R PfPDK1 mutation, the multiplication rates of NF54/PKAc cOE M1/PDK1_mut parasites overexpressing PfPKAc-GFP (–GlcN) reached only 60% to 80% compared to the matching control (+GlcN) (Figs 5D and S11). This observation suggested that an additional selection step had taken place in NF54/PKAc cOE S1 parasites that conferred full tolerance to PfPKAc-GFP OE (see Fig 3G). Indeed, based on the WGS data the parental NF54/PKAc cOE M1 clone carries 8 copies of the PfPKAc cOE transgene cassette, whereas the NF54/PKAc cOE S1 survivor population carries only 2 copies (S12 Fig). We therefore believe that due to the negative impact of continuous PfPKAc-GFP OE on parasite viability, NF54/PfPKAc cOE S1 parasites carrying fewer PfPKAc cOE cassettes and hence lower overall PfPKAc expression levels had a comparative advantage during the selection process for PfPKAc-GFP OE tolerance.

These data collectively confirm the importance of the *pfpdk1* mutations identified in NF54/PKAc cOE survivor parasites in conferring resistance to PfPKAc OE and suggest that PfPDK1 is the kinase that phosphorylates and activates PfPKAc. They furthermore imply that the PfPDK1 M51R mutant kinase can still phosphorylate PfPKAc with reduced efficiency and therefore sustain parasite viability in the presence of elevated PfPKAc levels.

### The PfPKAc activation loop residue T189 is essential for PfPKAc activity

Previous research in human cell lines and yeast identified a specific threonine residue in the PKAc activation loop (T197 in mammals) as the target of PDK1-dependent phosphorylation [15,18,23]. In PfPKAc, T189 likely represents the activation loop phosphorylation site corresponding to T197 in human PKAc. Hence, we tested whether the T189 residue is indeed important for PfPKAc activity. To achieve this, we employed the same approach as already used to obtain NF54/PKAc cOE parasites (S5 Fig) to generate the NF54/PKAcT189V cOE line that conditionally overexpresses a mutated version of PfPKAc in which T189 has been

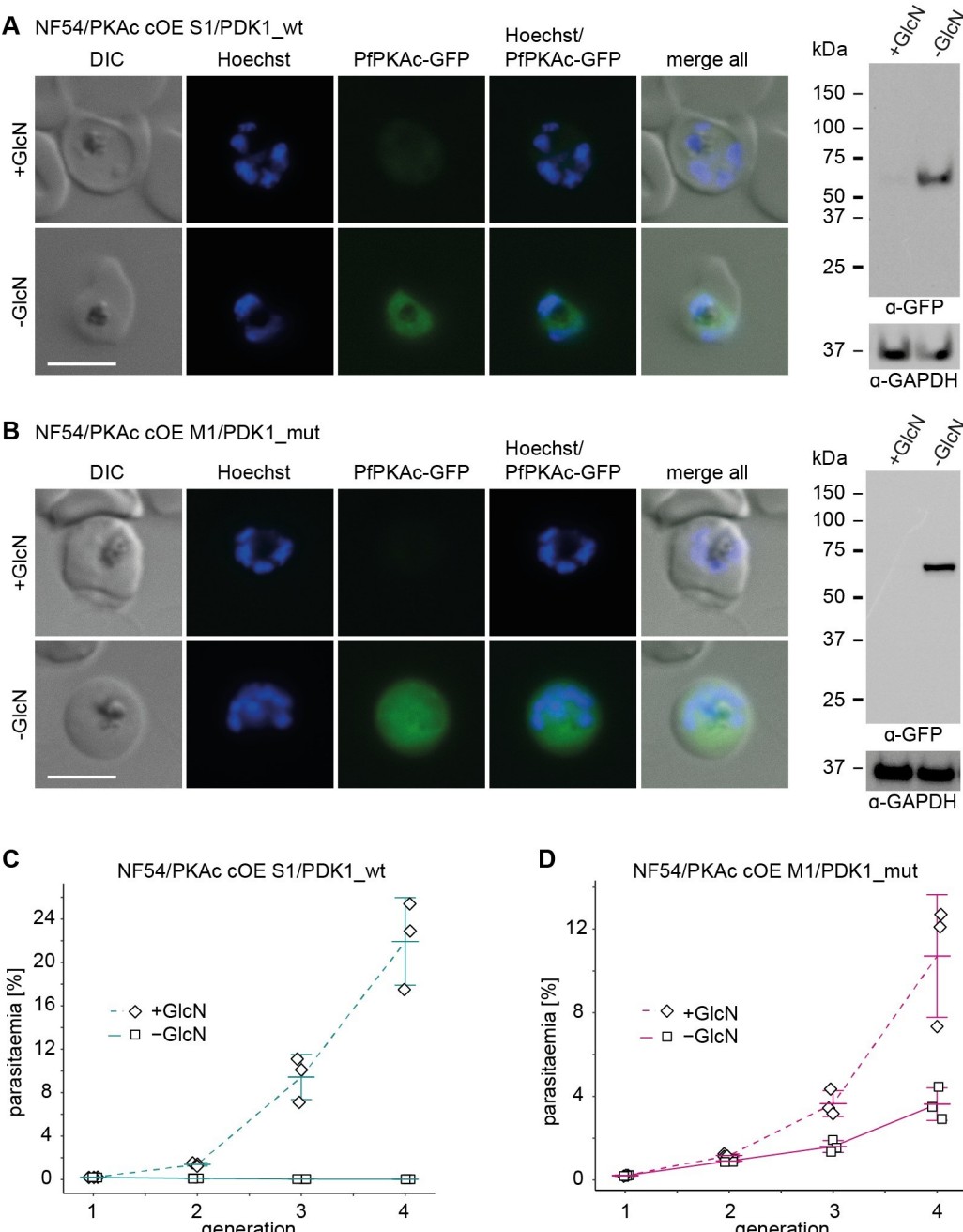

**Fig 5. Targeted mutagenesis of PfPDK1 confirms its essential role in activating PfPKAc. (A, B)** Expression of PfPKAc-GFP under OE-inducing (–GlcN) and control conditions (+GlcN) in NF54/PKAc cOE S1/PDK1_wt parasites (A) and NF54/PKAc cOE M1/PDK1_mut parasites (B) by live cell fluorescence imaging and western blot analysis. Synchronous parasites (0 to 8 hpi) were split (±GlcN) 40 hours before collection of the samples. Representative fluorescence images are shown. Parasite DNA was stained with Hoechst. Scale bar = 5 µm. For western blot analysis, lysates derived from an equal number of parasites were loaded per lane. MW PfPKAc-GFP = 67.3 kDa, MW PfGAPDH = 36.6 kDa. The full size western blots are shown in S11 Fig. **(C, D)** Increase in parasitaemia of NF54/PKAc cOE S1/PDK1_wt parasites (C) and NF54/PKAc cOE M1/PDK1_mut parasites (D) over 3 generations under PfPKAc-GFP OE-inducing (–GlcN) and control conditions (+GlcN). Synchronous parasites (0 to 6 hpi) were split (±GlcN) 18 hours before the first measurement in generation 1. Open squares represent data points for individual replicates and the means and SD (error bars) of 3 biological replicates are shown. The raw data are available in the source data file (S2 Data). DIC, differential interference contrast; GlcN, glucosamine.

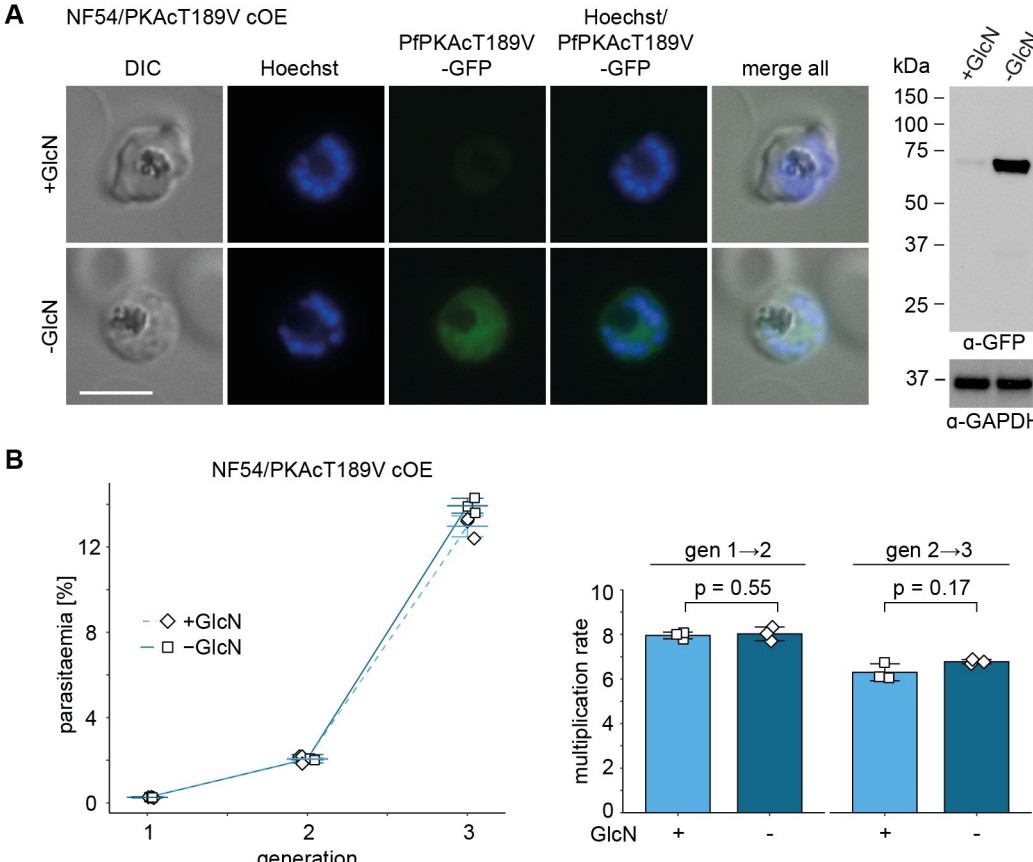

**Fig 6. Overexpression of PfPKAcT189V has no effect on intraerythrocytic parasite development and multiplication.**
**(A)** Expression of PfPKAcT189V-GFP under OE-inducing (−GlcN) and control conditions (+GlcN) by live cell fluorescence imaging and western blot analysis. Synchronous parasites (0 to 8 hpi) were split (±GlcN) 40 hours before collection of the samples. Representative fluorescence images are shown. Parasite DNA was stained with Hoechst. Scale bar = 5 μm. GlcN, glucosamine. For western blot analysis, lysates derived from equal numbers of parasites were loaded per lane. MW PfPKAc-GFP = 67.3 kDa, MW PfGAPDH = 36.6 kDa. The full size western blot is shown in S13 Fig. **(B)** Increase in parasitaemia (left) and parasite multiplication rates (right) of NF54/PKAcT189V cOE parasites over 3 generations under PfPKAcT189V-GFP OE-inducing (−GlcN) and control conditions (+GlcN). Synchronous parasites (0 to 6 hpi) were split (±GlcN) 18 hours before the first measurement in generation 1. Open squares represent data points for individual replicates and the means and SD (error bars) of 3 biological replicates are shown. Differences in multiplication rates have been compared using a paired 2-tailed Student $t$ test (statistical significance cutoff: $p < 0.05$). The raw data are available in the source data file (S2 Data). DIC, differential interference contrast.

substituted with a nonphosphorylatable valine residue [58,59] (S13 Fig). Correct insertion of the PfPKAcT189V cOE cassette into the *glp3* locus was verified by PCR on gDNA (S13 Fig). Live cell fluorescence microscopy and western blot analysis confirmed that PfPKAcT189V-GFP OE was efficiently induced upon removal of GlcN (Figs 6A and S13). Strikingly, in contrast to the lethal effect provoked by OE of PfPKAc-GFP, OE of the PfPKAcT189V-GFP mutant (−GlcN) had no effect on intraerythrocytic parasite development, multiplication, and survival when compared to the control population (+GlcN) (Fig 6B). Hence, these results suggest that PfPKAc activity is strictly dependent on phosphorylation of T189 in the activation loop segment. In combination with the findings obtained through the mutational analysis of PfPDK1, they further imply that PfPDK1 is directly or indirectly responsible for T189 phosphorylation and thus activation of PfPKAc.

## Kinase inhibitors targeting human PDK1 are active against asexual blood stage parasites

Many PDK1-dependent AGC kinases (e.g., AKT/PKB, RSK, PKC, S6K, and SGK) are downstream effectors in the PI3K/protein kinase B (AKT) or mitogen-activated protein kinase (MAPK) growth factor signalling pathways and are aberrantly activated in various types of cancer in humans [60]. In addition, PDK1 expression itself is augmented in many tumours [60]. For these reasons, human PDK1 is pursued as a potential drug target for cancer therapy, and a large number of inhibitors targeting human PDK1 have been developed and patented over the past 15 years [61–63]. For instance, BX-795 and BX-912, 2 related aminopyrimidine compounds, inhibit recombinant human PDK1 activity with half-maximal inhibitory concentrations ($IC_{50}$) of 11 nM and 26 nM, respectively [64], and the aminopyrimidine-aminoindazole GSK2334470 has similar in vitro potency against PDK1 (15 nM $IC_{50}$) [65,66]. While BX-795 was shown to also inhibit several other human kinases in vitro [67], GSK2334470 displayed high specificity for PDK1 over a large panel of other recombinant human kinases [65]. In cell-based assays, all 3 compounds inhibited the PDK1-dependent phosphorylation of several AGC kinase substrates at submicromolar concentrations [64–66]. Here, we tested these 3 commercially available ATP-competitive inhibitors of human PDK1 for their potential to kill *P. falciparum* asexual blood stage parasites using a [$^3$H]-hypoxanthine incorporation assay [68]. We found that BX-795, BX-912, and GSK2334470 all inhibited parasite proliferation with $IC_{50}$ values of 1.83 μM (± 0.23 SD), 1.31 μM (± 0.24 SD), and 1.83 μM (± 0.11 SD), respectively (S14 Fig). In an attempt to test whether the lethal effect of these molecules is due to the specific inhibition of PfPDK1, we repeated the dose response assays on NF54/PDK1 cKD parasites cultured in the presence (control) or absence of Shield-1 (PfPDK1 depleted). However, we observed no reduction in $IC_{50}$ values for PfPDK1 depleted (−Shield-1) compared to NF54/PDK1 cKD control parasites (+Shield-1) or compared to NF54 WT parasites cultured in the presence or absence of Shield-1 (S14 Fig), suggesting that all 3 inhibitors are not specific for PfPDK1 but likely target additional/other essential parasite kinases. To investigate the effects these compounds have on intraerythrocytic parasite development, we assessed the morphology of drug-treated parasites by visual inspection of Giemsa-stained thin blood smears. Analysis of NF54 WT early ring stage parasites treated with each of the 3 PDK1 inhibitors over a period of 60 hours revealed that parasites were unable to progress beyond the ring stage and became pyknotic thereafter, in contrast to untreated control parasites that progressed through schizogony and gave rise to ring stage progeny as expected (S14 Fig). Furthermore, additional results obtained from separate 18-hour treatments of early ring stages, early trophozoites or early schizonts suggest that all 3 inhibitors are active against all intraerythrocytic stages, as in all cases the morphology of drug-treated parasites was reminiscent of dying or pyknotic forms and viable trophozoites, schizonts, and ring stage progeny were not observed (S14 Fig). Hence, based on the promising activities of these molecules against blood stage parasites, attempts to identify their target(s) as well as the screening of extended PDK1 inhibitor libraries and experimental validation of PfPDK1 as a drug target would be worthwhile activities to be pursued in future research.

## Discussion

PfPKA is essential for the proliferation of asexual blood stage parasites, and the phosphorylation of the invasion ligand AMA1 is one of its crucial functions [5,6,8,9]. In addition, recent research described important roles for the PDE PfPDEβ and the AC PfACβ in regulating cAMP levels and hence PfPKA activity [8,27]. Here, we studied the function and regulation of the catalytic PfPKA subunit PfPKAc to obtain further insight into PfPKA-dependent signalling in *P. falciparum* blood stage parasites.

Using the NF54/AP2-G-mScarlet/PKAc cKD parasite line, we confirmed the previously described essential role for PfPKAc in merozoite invasion [8,9]. We also demonstrated that PfPKAc plays no major role in regulating sexual commitment. The dispensability of PfPKAc in the sexual commitment pathway seems rather surprising since early studies performed over 3 decades ago claimed a potential involvement of cAMP signalling in regulating sexual commitment [69,70]. However, these studies only indirectly suggested an involvement of cAMP/PfPKA-signalling in this process. For instance, Kaushal and colleagues determined the effect of high exogenous cAMP concentrations (1 mM) on sexual commitment and reported that under static culture conditions (high parasitaemia without addition of uninfected red blood cells [uRBCs]) nearly all parasites developed into gametocytes [69]. Rather than reflecting the true induction of sexual commitment by cAMP signalling, we suspect that observations may have been related to the selective killing of asexual stages by high cAMP concentrations (as indeed reported in their study) and/or the stimulation of high SCRs due to LysoPC depletion from the growth medium at high parasitaemia [3]. Our results further suggest that PfPKAc is not required for the morphological maturation of gametocytes and for male gametogenesis. However, even though our cKD system allowed for efficient depletion of PfPKAc expression, we cannot exclude the possibility that residual PfPKAc expression levels still supported normal sexual development. At this point, we would also like to reiterate that our experiments conducted with the NF54/AP2-G-mScarlet/PKAc cKD line showed that GlcN (2.5 mM), but not Shield-1 (675 nM), acts as a confounding factor when studying sexual commitment and male gametocyte exflagellation. We therefore advise to use the FKBP/DD-Shield-1 [36,37] or DiCre/rapamycin [71–73] conditional expression systems when studying these processes.

The cellular rigidity of immature *P. falciparum* gametocytes is linked to a dynamic reorganisation of the RBC spectrin and actin networks [45] as well as the presence of parasite-encoded STEVOR proteins at the iRBC membrane [47,74]. By contrast, the increased deformability gained by stage V gametocytes is accompanied by the reversal of these cytoskeletal rearrangements [45] and dissociation of STEVOR from the iRBC membrane [47,74]. Interestingly, results obtained from experiments using pharmacological agents to increase cellular cAMP levels or to inhibit PKA activity demonstrated that gametocyte rigidity is positively regulated by cAMP/PKA-dependent signalling [30,74]. While potential PKA substrates involved in this process are largely unknown, the PKA-dependent phosphorylation of the cytoplasmic tail of STEVOR (S324) is important to maintain cellular rigidity of immature gametocytes, and dephosphorylation of this residue is linked to the increased deformability of stage V gametocytes [74]. Given that PfPKA is not known to be exported into the iRBC cytosol, however, PKA-dependent phosphorylation events in the RBC compartment are likely exerted by human rather than parasite PKA. Notably though, overexpression of the regulatory subunit PfPKAr, which is expected to lower PfPKAc activity, caused increased deformability of stage III gametocytes [30]. Consistent with these data, we demonstrated that PfPKAc depletion caused a significant, yet only moderate, increase in stage III and V gametocyte deformability. While these results provide direct evidence for a role of PfPKAc-dependent phosphorylation in regulating the biomechanical properties of gametocyte-iRBCs, they also suggest that cAMP signalling through PfPKA is not the only driver of this process. We envisage that PfPKAc activity may regulate the expression, trafficking, or function of parasite-encoded proteins destined for export into the iRBC or of proteins of the inner membrane complex and/or microtubular and actin networks underneath the parasite plasma membrane that play important roles in determining cellular shape during gametocytogenesis [75–77]. Comparative phosphoproteomic analyses of the conditional PfPKAc mutants generated here and elsewhere [8,9] may be a promising approach to test this hypothesis and identify the actual substrates involved.

Three recent studies employing DiCre-inducible KO parasites for PfPKAc [8,9], the PDE PfPDEβ [27], and the AC PfACβ [8] highlighted the importance for tight regulation of PfPKAc activity in asexual blood stage parasites. In these studies, induction of the corresponding gene KOs in ring stage parasites caused no immediate defects in intraerythrocytic parasite development but resulted in a complete or severe block in RBC invasion by newly released merozoites due to prevention of PfPKAc activity (PfPKAc and PfACβ KOs) [8,9] or PfPKAc hyperactivation (PfPDEβ KO) [27], respectively. These findings are consistent with the specific expression pattern of all 3 cAMP signalling components in late schizont stages [78]. Interestingly, however, Flueck and colleagues showed that some PfPDEβ KO merozoites successfully invaded RBCs but were then unable to develop into ring stage parasites [27], providing compelling evidence for a lethal effect of PfPKAc hyperactivity also on early intraerythrocytic parasite development. Similarly, we showed that the OE of PfPKAc through a constitutively active heterologous promoter blocked parasite progression through schizogony. We believe this detrimental effect is due to the incapacity of endogenous PfPKAr to complex and inactivate the excess of PfPKAc enzymes, resulting in illegitimate activity of free PfPKAc and hence untimely phosphorylation of substrates prior to the intrinsic PfPKA activity window in late schizonts. While we did not engage in further explorations towards identifying the molecular mechanisms underlying the lethal consequences of PfPKAc overexpression, we discovered another kinase that is likely required for PfPKAc activation. We identified this function by selecting for "PfPKAc OE survivor" parasites able to tolerate PfPKAc OE. All 6 independently selected NF54/PKAc cOE survivor populations carried mutations in the same gene encoding a putative serine/threonine kinase (Pf3D7_1121900). Bioinformatic analyses and structural modelling suggested this kinase is an orthologue of the eukaryotic phosphoinositide-dependent protein kinase 1 (PDK1), hence termed PfPDK1.

Interestingly, all PfPDK1 mutations identified in the various NF54/PKAc cOE survivors are positioned proximal to the ATP-binding cleft and do not coincide with the PIF-binding pocket, suggesting that these mutations impair the catalytic efficiency of PfPDK1 rather than its capacity to interact with substrates. It therefore seems that the most straightforward manner for the parasite to overcome the lethal effect of PfPKAc OE was to acquire mutations reducing PfPKAc activation through PfPDK1-mediated phosphorylation. We confirmed this scenario by (1) reverting the PfPDK1 M51R mutation in the NF54/PKAc cOE S1 survivor, which rendered these parasites again sensitive to PfPKAc OE; and (2) by introducing the PfPDK1 M51R mutation into the PfPKAc OE-sensitive clone M1, which rendered these parasites resistant to PfPKAc OE. Notably, we could also show that OE of the PfPKAcT189V mutant form of PfPKAc, which carries a non-phosphorylatable valine residue instead of the target threonine in the activation loop, had no negative effect on intraerythrocytic parasite development and multiplication. Together, these striking results imply that in addition to the cAMP-mediated release of PfPKAc from the regulatory subunit PfPKAr, phosphorylation of the T189 residue is essential for PfPKAc activity and provide compelling evidence that PfPDK1 is the kinase that targets this residue. While this hypothesis is entirely consistent with the evolutionary conserved role for PDK1 in activating PfPKAc in other eukaryotes [14–18], further experiments will be required to confirm that PfPDK1 indeed interacts with and activates endogenous PfPKAc via T189 phosphorylation in vivo.

Conditional depletion of PfPDK1 did not result in any obvious multiplication or developmental defects in asexual and sexual blood stage parasites, showing that largely diminished PfPDK1 protein levels are still sufficient to sustain parasite viability and proliferation. However, several lines of evidence strongly argue for an essential role for PfPDK1 in asexual parasites and that at least one of its vital functions is to activate PfPKAc. First, previous studies [31,54] and our own attempts failed to obtain PfPDK1 null mutants via gene disruption approaches. Second, none of the different mutations identified in the 6 NF54/PKAc cOE

survivors introduced a nonsense loss-of-function mutation into the *pfpdk1* open reading frame. This observation again supports the notion that PfPDK1 function is vital and that the PfPDK1 mutant enzymes retain residual kinase activity. Third, we were only able to introduce the M51R PfPDK1 mutation into parasites overexpressing PfPKAc but not into WT parasites, suggesting that parasites expressing functionally compromised PfPDK1 mutants can only survive if impaired PfPDK1-dependent PfPKAc activation is compensated for by elevated PfPKAc expression levels. Given that PDK1 is widely conserved in eukaryotes and required to activate AGC kinases at large [17], it is conceivable that PfPDK1 may also regulate the activity of PfPKG or PfPKB, the only other 2 known members of the AGC family in *P. falciparum*, which are both essential in blood stage parasites [31,79]. Furthermore, the manifestation of the PfPKAc OE phenotype in late trophozoites/early schizonts implies that PfPDK1 is active and phosphorylates other substrates already at this stage, hours prior to the expression window of endogenous PfPKAc in late schizonts. Importantly, however, the fact that the NF54/PKAc cOE survivor parasites expressing functionally impaired PfPDK1 mutant enzymes are fully viable suggests that the PfPDK1-dependent phosphorylation of other substrates is either not essential or can still be executed at functionally relevant baseline levels by mutated PfPDK1.

In summary, we provide unprecedented functional insight into the cAMP/PfPKA signalling pathway in the malaria parasite *P. falciparum*. Our results complement earlier studies highlighting the importance of tight regulation of PfPKA activity for parasite survival, showing that diminished as well as augmented PfPKAc expression levels are lethal for asexual blood stage parasites. In addition to the well-established roles for the regulatory subunit PfPKAr, the AC PfACβ and the PDE PfPDEβ in regulating PfPKAc activity via cAMP levels, we provide compelling evidence that PfPDK1 is required to activate PfPKAc, most likely through activation loop phosphorylation at T189. In light of the essential role for PfPDK1 in this and possibly other parasite AGC kinase-dependent signalling pathways, and the promising anti-parasite activity of PDK1 kinase inhibitors, PfPDK1 represents an attractive candidate for further functional and structural studies and to be explored as a possible new antimalarial drug target.

## Materials and methods

### Parasite culture

*P. falciparum* NF54 parasites were cultured and asexual growth was synchronised using 5% sorbitol as described previously [80,81]. Parasites were cultured in AB+ or B+ human RBCs (Blood Donation Center, Zurich, Switzerland) at a haematocrit of 5%. The standard parasite culture medium (PCM) contains 10.44 g/L RPMI-1640, 25 mM HEPES, 100 μM hypoxanthine and is complemented with 24 mM sodium bicarbonate and 0.5% AlbuMAX II (Gibco #11021–037). Moreover, 2 mM choline chloride (CC) was routinely added to PCM to block induction of sexual commitment [3]. To induce sexual commitment, parasites were cultured in serum-free medium (–SerM) as described previously [3].–SerM medium contains fatty acid-free BSA (0.39%, Sigma #A6003) instead of AlbuMAX II and 30 μM oleic and 30 μM palmitic acid (Sigma #O1008 and #P0500) [3]. SCR assays were performed using standardised–SerM medium complemented with 2 mM CC (–SerM/CC) [3]. Gametocytes were cultured in PCM complemented with 10% human serum (Blood Donation Center, Basel, Switzerland) instead of AlbuMAX II (+SerM). Parasite cultures were kept in gassed (4% $CO_2$, 3% $O_2$, 93% $N_2$) airtight containers at 37˚C.

### Transfection constructs

The NF54/AP2-G-mScarlet/PKAc cKD parasite line was generated using a SLI-based gene editing approach [35]. For this purpose, the SLI_PKAc_cKD transfection construct was

generated in 3 successive cloning steps using Gibson assembly reactions. First, the SLI_cKD precursor plasmid was generated by joining 4 fragments in a Gibson assembly reaction: (1) the plasmid backbone amplified from pUC19 (primers PCRA_F and PCRA_R) as previously described [53]; (2) the *SpeI/BamHI* cloning site followed by the *gfp-dd* sequence amplified from pD_*ck2α-gfpdd* [82] using primers gfp_F and dd_R; (3) the *2a-bsd* sequence amplified from pSLI-BSD [35] (primers 2A_F and bsd_R); and (4) the *glmS-hrp2* 3′ sequence amplified from pD_cOE (described below) (primers glmS1_F and term_R). Second, a 4-fragment Gibson assembly reaction was performed using (1) the *SalI*- and *EcoRI*-digested SLI_cKD precursor plasmid; (2) the *calmodulin* (PF3D7_1434200) promoter amplified from pHcamGFP-DD [53] (primers cam1_F and cam1_R); (3) the yeast dihydroorotate dehydrogenase (y*dhodh*) resistance gene (conferring resistance to DSM1) amplified from the pUF1-Cas-9 plasmid [83] (primers ydhodh_F and ydhodh_R); and (4) the *pbdt 3′* terminator amplified from pUF1-Cas-9 [83] (primers pbdt3_F and pbdt3_R). Third, this SLI_cKD_ydhodh precursor plasmid was digested using *SpeI* and *BamHI* and joined with the 3′ HR of *pfpkac* amplified from NF54 gDNA (primers pka3′_F and pka3′_R) in a 2-fragment Gibson assembly reaction resulting in the final SLI_PKAc_cKD transfection construct.

The CRISPR/Cas-9 gene editing system employed here is based on cotransfection of suicide and donor plasmids [53]. The pBF-gC- or pHF-gC-derived suicide plasmids encode the Cas-9 enzyme, the single guide RNA (sgRNA) cassette, and either the blasticidin deaminase (BSD; conferring resistance to BSD-S-HCl) or the human dihydrofolate reductase (hDHFR; conferring resistance to WR99210) resistance markers fused to the negative selection marker yeast cytosine deaminase/uridyl phosphoribosyl transferase (BSD-yFCU or hDHFR-yFCU) [53]. The pD-derived donor plasmid [53] contains the sequence assembly essential for homology-directed repair of the DNA double-strand break induced by Cas-9. To obtain the NF54/PKAc cOE parasite line, the pD_pkac_cOE donor plasmid was generated in several cloning steps using Gibson assembly reactions. First, a pD_cOE_DD precursor plasmid was generated using a 2-fragment Gibson assembly joining (1) the plasmid backbone including the *glp3*-specific 5′ and 3′ HRs amplified from pD_*cg6_cam-gdv1-gfp-glmS* [84] (primers glp3_F and glp3_R); and (2) the *cam 5′-gfpdd-hrp2 3′* sequence amplified from pHcamGFP-DD [53] (primers cam_F and hrp2_R). Second, the subsequent precursor plasmid pD_cOE_glmS was generated by performing a Gibson assembly using 2 fragments (1) the pD_cOE_DD precursor plasmid digested using *SalI* and *AgeI*; and the (2) *glmS* sequence amplified from pD_*cg6_cam-gdv1-gfp-glmS* [84] (primers glmS_F and glmS_R). To insert a new cloning site and generate the next precursor plasmid pD_cOE, another Gibson assembly reaction was performed joining (1) the pD_cOE_glmS plasmid digested using *BamHI* and *NotI*; and (2) annealed complementary oligonucleotides clon_F and clon_R. The final pD_pkac_cOE plasmid was generated by assembling 2 Gibson fragments: (1) the *BamHI*- and *SpeI*-digested pD_cOE plasmid; and (2) the *pfpkac* sequence amplified from NF54 gDNA using primers pka_F and pka_R. The pBF_gC-*cg6* suicide plasmid that was cotransfected with pD_pkac_cOE to generate this NF54/PKAc cOE parasite line encodes the sgRNA targeting the *glp3* locus [84]. To obtain the NF54/PKAcT189V cOE parasite line, the pD_pkacT189V_cOE donor plasmid was generated in a Gibson assembly joining 3 fragments: (1) the *BamHI*- and *SpeI*-digested pD_cOE plasmid; (2) the 5′ fragment of the *pfpkac* sequence containing the single point mutation resulting in the T189V amino acid change (primers pka_F and T189V_R); and (3) the 3′ fragment of the *pfpkac* sequence overlapping with the 5′ fragment and coding for the same amino acid change (primers T189V_F and pka_R). As described above, the *glp3*-specific suicide plasmid pBF_gC-*cg6* was used for cotransfection. The NF54/PDK1 cKD parasite line was obtained by cotransfection of pHF_gC_pdk1-gfpdd (encoding sgRNA_pdk1) and pD_pdk1-gfpdd. The previously published suicide mother plasmid pHF-gC [53] was used to insert the sgRNA sequence

targeting the 3′ end of *pfpdk1* (sgt_pdk1). For this purpose, complementary oligonucleotides were annealed and the resulting double-stranded fragment was ligated into the *BsaI*-digested pHF-gC plasmid using T4 DNA ligase generating the pHF_gC_pdk1-gfpdd suicide plasmid. The pD_pdk1-gfpdd plasmid was generated by performing a 4-fragment Gibson assembly joining (1) the plasmid backbone amplified from pUC19 (primers PCRA_F and PCRA_R) as previously described [53]; (2) the 5′ HR amplified from NF54 gDNA (primers hr1KD_F and hr1KD_R); (3) the *gfpdd* sequence (primers gfpdd_F and gfpdd_R) amplified from pHcamGDV1-GFP-DD [53]; and (4) the 3′ HR amplified from NF54 gDNA (hr2KD_F and hr2KD_R). The parasite lines NF54/PKAc cOE M1/PDK1_mut and NF54/PKAc cOE S1/PDK1_wt were generated as follows. Plasmids pHF_gC_S1rev and pHF_gC_M1mut were generated by inserting annealed complementary oligonucleotides encoding the respective sgRNAs (sgRNA_S1rev or sgRNA_M1mut) into the *BsaI*-digested pHF_gC suicide vector [53]. The donor plasmids pD_S1rev and pD_M1mut were generated in a 3-fragment Gibson assembly joining (1) the plasmid backbone amplified from pUC19 (primers PCRA_F and PCRA_R) as previously described [53]; (2) the corresponding 5′ HRs amplified from NF54 gDNA (primers hr1_F and either rev_R or mut_R); and (3) the respective 3′ HRs amplified from NF54 gDNA (primers hr2_R and either rev_F or mut_F). The primers rev_F/rev_R or mut_F/mut_R encode the WT (M) or mutated (R) amino acid number 51 of PfPDK1, respectively.

All primers used for cloning of the described transfection constructs are listed in S1 Table.

## Transfection and transgenic cell lines

*P. falciparum* ring stage parasite transfection was performed as described [53]. A total of 100 μg plasmid DNA was used to transfect NF54/AP2-G-mScarlet and NF54 WT parasites (100 μg of the SLI_PKAc_cKD plasmid; 50 μg each of all described CRISPR/Cas-9 suicide and donor plasmids). Moreover, 24 hours after transfection of the SLI_PKAc_cKD plasmid, parasites were cultured on 1.5 μM DSM1 until a stably growing parasite population was obtained. This culture was subsequently treated with 2.5 μg/mL BSD-S-HCl to select for parasites in which the *pfpkac* gene was successfully tagged. Similarly, 24 hours after transfection of CRISPR/Cas-9–based plasmids, the cultures were treated with 2.5 μg/mL BSD-S-HCl (for 10 subsequent days) or 4 nM WR99210 (for 6 subsequent days) depending on the resistance cassette encoded by the suicide plasmid. NF54/AP2-G-mScarlet/PKAc cKD and NF54/PDK1 cKD parasites were constantly cultured on 675 nM Shield-1 (+Shield-1) to stabilise the PfPKAc-GFPDD or PfPDK1-GFPDD protein, respectively. NF54/PKAc cOE, NF54/PKAcT189V cOE, NF54/PKAc cOE M1/PDK1_mut, and NF54/PKAc cOE S1/PDK1_wt parasites were constantly cultured on 2.5 mM GlcN to block OE of PfPKAc or PfPKAcT189V. About 2 to 3 weeks after transfection, stably growing parasite cultures were obtained and diagnostic PCRs on gDNA were used to confirm correct genome editing. Primers used to verify for correct gene editing are listed in S2 Table. Correct targeted mutagenesis of the *pfpdk1* gene in NF54/PKAc cOE M1/PDK1_mut and NF54/PKAc cOE S1/PDK1_wt parasites was confirmed by Sanger sequencing. Sanger sequencing data analysis and visualisation was performed using SnapGene software 4.1.6 (Insightful Science, San Diego, CA, USA).

## Limiting dilution cloning

Limiting dilution cloning was performed as previously described [49]. In brief, synchronous ring stage parasite cultures were diluted with fresh PCM and RBCs to a haematocrit of 0.75% and a parasitaemia of 0.0006% (= parasite cell suspension). Each well of a flat-bottom 96-well microplate (Costar #3596) was filled with 200 μL PCM/0.75% haematocrit (= RBC suspension). In each well of row A, 100 μL of the parasite cell suspension was mixed with the 200 μL

RBC suspension (1/3 dilution), resulting in a parasitaemia of 0.0002%, which equals approximately 30 parasites per well. Subsequently, 100 μL of the row A parasite cell suspensions were mixed with 200 μL RBC suspension in the wells of row B resulting again in a 1/3 dilution (approximately 10 parasites/well). This serial dilution was continued until the last row of the plate was reached. The 96-well microplate was kept in a gassed airtight container at 37°C for 11 to 14 days without medium change. Subsequently, using the Perfection V750 Pro scanner (Epson, Nagano, Japan), the 96-well microplate was imaged to visualise plaques in the RBC layer. The content of wells containing a single plaque was then transferred individually into 5-mL cell culture plates and cultured using PCM until a stably growing parasite culture was obtained.

## Flow cytometry

Quantification of parasite multiplication was performed using flow cytometry measurements of fluorescence intensity. For this purpose, synchronous parasites cultures (0.2% parasitaemia) were split at 0 to 6 hpi and cultured separately for the entire duration of the multiplication assay under (i) +Shield-1/−GlcN and−Shield-1/+GlcN conditions (NF54/AP2-G-mScarlet/PKAc cKD); (ii)−GlcN and +GlcN conditions (NF54/PKAc cOE M1, NF54/PKAc cOE S1, NF54/PKAc cOE S1/PDK1_wt, NF54/PKAc cOE M1/PDK1_mut, NF54/PKAcT189V cOE, NF54 WT); or (iii) +Shield-1 and−Shield-1 conditions (NF54/PDK1 cKD, NF54 WT). To determine the exact starting parasitaemia on day 1 of the assay (generation 1), gDNA of synchronous ring stage parasites (18 to 24 hpi) was stained for 30 minutes using SYBR Green DNA stain (1:10,000) (Thermo Fisher Scientific/Invitrogen, Reinach, Switzerland) at 37°C. Subsequently, parasites were washed twice in PBS, and fluorescence intensity was measured using the MACS Quant Analyzer 10. A total of 200,000 RBCs were measured per sample. The measurement was repeated for 2 (days 3 and 5) or 3 (days 3, 5, and 7) subsequent generations. The FlowJo_v10.6.1 software was used to analyse the flow cytometry data. The measurements were gated to remove small debris (smaller than cell size) and doublets (2 cells in a single measurement). Using an uRBC control sample, uRBCs were separated from iRBCs based on their SYBR Green intensity. Representative plots showing the gating strategy used for all flow cytometry data are presented in S2 Fig.

## Fluorescence microscopy

Live cell fluorescence imaging was performed to visualise protein expression as described [85]. Parasite nuclei were stained using 5 μg/ml Hoechst (Merck, Buchs, Switzerland) and Vectashield (Vector Laboratories, Burlingame, CA, USA) was used to mount the microscopy slides. Live cell fluorescence microscopy was performed using a Leica DM5000 B fluorescence microscope (20×, 40×, and 63× objectives), and images were acquired using the Leica application suite (LAS) software Version 4.9.0 and the Leica DFC345 FX camera. Images were processed using Adobe Photoshop CC 2018, and for each experiment identical settings for both image, acquisition and processing were used for all samples analysed.

For the quantification of the number of nuclei per schizont, synchronous NF54/PfPKAc cOE M1 and S1 ring stage parasites (0 to 6 hpi) were split and cultured either in presence or absence GlcN. Moreover, 40 hours later (40 to 46 hpi), schizonts were stained using 5 μg/ml Hoechst and Vectashield-mounted slides visually inspected using the Leica fluorescence microscope and the LAS software. Three biological replicate experiments were performed and for each replicate the number of nuclei in 100 schizonts was determined by manual counting.

## Western blot analysis

Parasite pellets were obtained by lysing RBCs using 0.15% saponin in PBS (10 mon ice) followed by centrifugation and washed in ice-cold PBS until the supernatant was clear. Whole parasite protein lysates were generated by solubilising the parasite pellet in an UREA/SDS buffer (8 M Urea, 5% SDS, 50 mM Bis-Tris, 2 mM EDTA, 25 mM HCl, pH 6.5) complemented with 1x protease inhibitor cocktail (Merck) and 1 mM DTT. Protein lysates were separated on NuPage 5–12% Bis-Tris or 3–8% Tris-Acetate gels (Thermo Fisher Scientific/Invitrogen, Reinach, Switzerland) using MES running buffer (Novex, Qiagen). Following protein transfer to a nitrocellulose membrane, the membrane was blocked for 30 minutes using 5% milk in PBS/0.1% Tween (PBS/Tween). Primary antibodies mouse mAb α-GFP (1:1,000) (#11814460001) (Roche Diagnostics, Rotkreuz, Switzerland) or mAb α-PfGAPDH [86] (1:20,000) diluted in blocking buffer were used for detection of GFP-tagged proteins or the GAPDH loading control, respectively. Primary antibody incubation was performed at 4˚C overnight in PBS/Tween, and the membrane was subsequently washed 3 times. Incubation using the secondary antibody α-mouse IgG (H&L)-HRP (1:10,000) (GE healthcare #NXA931) diluted in blocking buffer was performed for 2 hours. The membrane was washed 3 times using PBS/Tween before chemiluminescent signal detection using KPL LumiGLO Chemiluminescent Substrate System (SeraCare, Milford, MA, USA).

## Quantification of sexual commitment rates

High content imaging was performed to quantify SCRs of NF54/AP2-G-mScarlet/PKAc cKD and NF54/AP2-G-mScarlet control parasites. Synchronous parasites (0 to 6 hpi) were split (±Shield-1/±GlcN or ±GlcN), and 18 hours later (18 to 24 hpi), the culture medium was replaced with standardised–SerM/CC medium (2% parasitaemia, 2.5% haematocrit). Twenty-two hours later (sexually committed schizonts) or 48 hours later (sexually committed ring stage progeny), 30 μL of culture suspension were mixed with 50 μL Hoechst/PBS (8.1 μM) and incubated in a 96-well plate for 30 minutes. Cells were pelleted (300 g, 5 minutes) and washed twice using 200 μL PBS. Subsequently, the cell pellet was resuspended using 180 μL PBS and 30 μL suspension was pipetted into the wells of a clear-bottom 96-well plate (Greiner CELL-COAT microplate 655948, Poly-D-Lysine, flat μClear bottom) containing 150 μL PBS per well. Prior to imaging, cells were allowed to settle for 30 minutes at 37˚C. The MetaXpress software (version 6.5.4.532) (Molecular Devices, San Jose, CA, USA), the ImageXpress Micro XLS wide-field high content imaging system using a Plan-Apochromat 40x objective, and Sola SE solid state white light engine (Lumencor, Beaverton, OR, USA) were used for automated image acquisition and data analysis. The Hoechst (Ex: 377/50 nm, Em: 447/60 nm, 80 ms exposure) and mScarlet (Ex: 543/22 nm, Em: 593/40 nm, 600 ms exposure) filter sets were used for imaging all iRBCs and AP2-G-mScarlet-expressing parasites, respectively. Thirty-six sites per well were imaged. Quantification of both Hoechst-positive and AP2-G-mScarlet-positive parasites allowed calculating SCRs (percentage of AP2-G-mScarlet-positive parasites among all Hoechst-positive parasites).

To quantify SCRs of NF54/PDK1 cKD parasites, GlcNAc assays were performed [44]. For this purpose, synchronous parasites (0 to 6 hpi) were split (±Shield-1) and 18 hours later (18 to 24 hpi) the culture medium was replaced with standardised–SerM/CC medium (2% parasitaemia, 5% haematocrit). Upon reinvasion, the ring stage parasitaemia was quantified from Giemsa-stained thin blood smears prepared at 18 to 24 hpi. This parasitaemia corresponds to the cumulative counts of asexual ring stages and sexual ring stages (day 1 of gametocytogenesis). From 24 to 30 hpi onwards, parasites were cultured in +SerM medium supplemented with 50 mM GlcNAc (Sigma) to eliminate asexual parasites [44]. On day 4 of

gametocytogenesis (stage II gametocytes), the parasitaemia was again quantified from Giemsa-stained thin blood smears. SCRs were determined as the percentage of the day 4 parasitaemia (stage II gametocytes) compared the total parasitaemia observed on day 1.

## Gametocyte cultures

Synchronous gametocyte cultures were used to study gametocyte morphology, to extract protein samples and to perform microsphiltration and exflagellation assays. Sexual commitment was induced at 18 to 24 hpi using–SerM medium. Upon reinvasion (0 to 6 hpi) (asexual and sexual ring stages; day 1 of gametocytogenesis), parasites were cultured in +SerM medium. Another 24 hours later (24 to 30 hpi) (trophozoites and stage I gametocytes, day 2 of gametocytogenesis), 50 mM GlcNAc was added (+SerM/GlcNAc) to eliminate asexual parasites [44]. The +SerM/GlcNAc medium was changed daily for 6 consecutive days and thereafter gametocytes were cultured in +SerM medium that was replaced daily on a 37°C heating plate to prevent gametocyte activation.

## Microsphiltration experiments

Synchronous NF54/AP2-G-mScarlet/PKAc cKD and NF54 WT parasites were split at 0 to 6 hpi and cultured separately in presence and absence of Shield-1 (±Shield-1). Subsequently, sexual commitment was induced at 18 to 24 hpi using–SerM medium, and after reinvasion, gametocytes were cultured using +SerM/GlcNAc and +SerM medium as described above. On day 6 (stage III) and 11 (mature stage V) of gametocytogenesis, microsphiltration experiments were conducted as described previously [46]. Per sample and condition, either 1 (NF54 WT) or 2 (NF54/AP2-G-mScarlet/PKAc cKD) independent biological replicate experiments with 6 technical replicates each were conducted. The experiment starts by transferring gametocyte culture aliquots into 15 mL Falcon tubes and lowering the haematocrit to 1.5% by addition of PCM. Six microsphere filters (technical replicates) were loaded per sample and condition. After injection of 600 μL cell suspension, filters were washed with 5 mL +SerM medium at a speed of 60 mL per hour using a medical grade pump (Syramed μSP6000, Acromed, Switzerland). Gametocytaemia before ("UP") and after ("DOWN") the microsphiltration process were determined from Giemsa-stained thin blood smears by counting at least 1,000 RBCs. The "UP" gametocytaemia was determined as the mean gametocytaemia calculated from 2 independent Giemsa-stained thin blood smears. The "DOWN" gametocytaemia was determined for each filter separately. Gametocyte retention rates were calculated as 1-("DOWN" gametocytaemia divided by "UP" gametocytaemia). Samples were kept at 37°C whenever possible to prevent gametocyte activation and lack thereof was confirmed by visual inspection of Giemsa-stained thin blood smears.

## Exflagellation assays

On day 14 of gametocytogenesis (mature stage V), exflagellation assays were performed as described previously [87]. Briefly, in a Neubauer chamber gametocytes were activated using 100 μM xanthurenic acid (XA) and a drop in temperature (from 37°C to 22°C). After 15 minutes of activation, the total number of RBCs per mL of culture and the number of exflagellation centres by activated male gametocytes were quantified by bright-field microscopy (40× objective). The gametocytaemia before activation was determined from Giemsa-stained thin blood smears. ExRs were calculated as the proportion of exflagellating gametocytes among all gametocytes. At least 3 biological replicates were performed per experiment.

## Illumina whole genome sequencing

To perform WGS, gDNA of the NF54/PKAc cOE clones M1 and M2 and the 6 independently grown NF54/PKAc cOE survivors (S1-S6) was isolated using a phenol/chloroform-based extraction protocol as described [88]. To avoid an amplification bias due to the high AT-content of *P. falciparum* gDNA, DNA sequencing libraries were prepared using the PCR-free KAPA HyperPrep Kit (Roche). Libraries were sequenced on an Illumina NextSeq 500 and the quality of the raw sequencing reads was analysed with FastQC (version 0.11.4) [89]. The raw reads were mapped to the *P. falciparum* 3D7 reference genome (PlasmoDB version 39) complemented with the corresponding transfection plasmid sequences using the Burrows-Wheeler Aligner (version 0.7.17) [90] with default parameters. The alignment files in SAM format were converted to binary BAM files with SAMtools (version 1.7) [91], and the BAM files were coordinate sorted and indexed, and read groups were added with Picard (version 2.6.0) [92]. Sequence variants (SNP and Indels) were directly called with the Genome Analysis Toolkit's (GATK, version 4.0.7.0) HaplotypeCaller in the GVCF mode to allow multisample analysis [93]. The resulting g.vcf files of the different samples were combined into one file and genotyped using GATK [93]. To predict the consequences of the obtained variants, they were annotated with SnpEff (version 4.3T) [94] using the SnpEff database supplemented manually with the 3D7 genome annotation (PlasmoDB version 39). The detected variants were filtered for (i) "HIGH" or "MODERATE" impact (thus nonsynonymous variants); (ii) absence in NF54/PKAc cOE clones M1 and M2; and (iii) an allele frequency of the alternative allele of >40% in at least one of the NF54/PKAc cOE survivors (S1-S6). The obtained list of 83 candidate variants was first screened manually (i) for variants present in all NF54/PKAc cOE survivors; and (ii) for genes mutated in all NF54/PKAc cOE survivors, leaving the variants identified in Pf3D7_1121900 as only candidates. Additionally, all 83 original candidate variants were inspected visually with the Integrative Genomics Viewer (version 2.7.0) [95]. Variants were excluded if they were suspected to be false positives because they were (i) only supported by a very small number of reads, and (iia) also detected in reads of the mother clones and not called because of low allele frequencies or (iib) insertions and deletions after/before large homopolymers or repeat tracts. This again left the variants in Pf3D7_1121900 as only candidates.

To analyse PfPKAc-GFP cOE cassette copy numbers, the sequencing coverage over the whole genome was determined in 50-nucleotide windows using the software igvtools (version 2.5.3) [95] and the mean coverage of (i) the whole genome; (ii) the endogenous *pfpkac* locus; and (iii) the ectopic *pfpkac-gfp* cOE cassettes were calculated in RStudio (R version 3.6.2, RStudio release 1.2.5033). The mean coverage of the endogenous *pfpkac* locus and ectopic *pfpkac-gfp* cOE cassettes was summed up, as both sequences are identical and reads derived from the ectopic *pfpkac-gfp* cOE cassettes were mapped to both alleles. *pfpkac* coverage was normalised to the mean genome-wide coverage (assuming the copy number of the genome is 1) and to the *pfpkac* coverage of WT parasites. Finally, the endogenous *pfpkac* was subtracted (−1) to obtain the approximate copy number of ectopic PfPKAc-GFP cOE cassettes.

## Sequence alignments and modelling of protein structure

Alignment of the PfPDK1 (Pf3D7_1121900/UniProt ID Q8IIE7) amino acid sequence with PDK1 from *P. vivax* (PVX_091715, UniProt ID A5K4N1), *Arabidopsis thaliana* (UniProt ID Q9XF67), *Caenorhabditis elegans* (UniProt ID Q9Y1J3), and humans (UniProt ID O15530) was performed using Clustal Omega [96]. A homology model of the PfPDK1 structure was built using SWISS-MODEL [97] based on the human PDK1 crystallographic structure (PDB ID 1UU9) [52]. The mean homology model quality (Global Model Quality Estimation, GMQE) was assessed as 0.49, suggesting a model of average quality (possible GMQE values are

0 to 1 on a linear scale, higher values indicate better quality). A large, predicted disordered loop (amino acids 180 to 310) present in PfPDK1 but not in its homologues was not built in the model. Amino acids of the ATP-binding cleft were defined as those within 4 Å of any ATP atom in the human PDK1 structure. The PIF-binding pocket was defined as suggested by Biondi and colleagues [22].

## Drug assays

Activity of human PDK1 kinase inhibitors on asexual NF54 WT and NF54/PDK1 cKD parasite multiplication was determined using an [$^3$H] hypoxanthine incorporation assay [68]. The mean $IC_{50}$ values were determined from 3 biological replicate assays, each performed in technical duplicate. BX-795 (CAS Number 702675-74-9, Selleckchem #S1274), BX-912 (CAS Number 702674-56-4, Selleckchem #S1275), and GSK2334470 (CAS Number 1227911-45-6, Selleckchem #S7087) were resuspended in DMSO and used at a maximum starting concentration of 50 μM, 10 μM, or 5 μM as indicated in the source data file (S2 Data). In a 6-step serial dilution, the compound concentration was diluted to half in each step to span a concentration range between 50 μM and 0.8 μM, 10 μM and 0.15 μM, or 5 μM and 0.075 μM, respectively. Chloroquine (CAS Number 50-63-5, Sigma #C6628) and artesunate (CAS Number 88495-63-0, Mepha #11665) served as control antimalarial compounds with a starting concentration of 100 nM and 50 nM, respectively. In a 6-step serial dilution, the chloroquine and artesunate concentrations were diluted to half in each step to span a concentration range between 100 nM and 1.5 nM or between 50 nM and 0.8 nM, respectively.

To investigate the effect of human PDK1 inhibitors on intraerythrocytic parasite development, synchronous parasite populations were exposed to BX-795, BX-912, or GSK2334470 at a concentration 10 times the $IC_{50}$ (10x $IC_{50}$) and parasite morphology was assessed on Giemsa-stained thin blood smears prepared at subsequent time points by bright-field light microscopy (100x objective). In a first assay, synchronous young ring stages (0 to 4 hpi) were exposed to the drugs for 60 hours and Giemsa-stained blood smears prepared every 12 hours. In a second assay, synchronous young ring stages (0 to 4 hpi), late ring stages/early trophozoites (20 to 24 hpi), or late trophozoites/early schizonts (30 to 34 hpi) were exposed to the drugs for 18 hours, after which the drugs were washed out by washing the cells in 2.5 volumes of culture medium and placing them back into culture using drug-free medium for another 24 hours. Giemsa-stained blood smears were prepared 18, 30, and 42 hours after the start of the assay.

## Statistical analysis

All data from assays quantifying parasite multiplication, number of nuclei per parasite, SCRs, gametocyte deformability, ExRs, and $IC_{50}$ values are represented as means with error bars defining the standard deviation. All data were derived from at least 3 biological replicate experiments. Statistical significance ($p < 0.05$) was determined using paired or unpaired Student $t$ tests as indicated in the figure legends. The exact number of biological replicates performed per experiment and the number of cells analysed per sample are indicated in the figure legends and in the source data file (S2 Data). Data were analysed and plotted using RStudio Version 1.1.456 and package ggplot2 or GraphPad Prism Version 8.2.1 for Windows (Insightful Science, San Diego, CA, USA).

## Supporting information

**S1 Fig. SLI-based engineering and characterisation of the NF54/AP2-G-mScarlet/PKAc cKD parasite line. (A)** Top: Scheme depicting the WT *pfpkac* locus, the SLI_PKAc_cKD construct transfected into NF54/AP2-G-mScarlet parasites and the edited *pfpkac* locus in NF54/

AP2-G-mScarlet/PKAc cKD parasites. Primers used for diagnostic PCRs are indicated. Middle: The schematic map of the edited *pfap2-g-mScarlet* locus in the NF54/AP2-G-mScarlet parasite line [41] is shown in brackets. Bottom: Results of PCR reactions performed on gDNA of NF54/AP2-G-mScarlet/PKAc cKD and NF54 WT control parasites confirm correct gene editing. **(B)** Full size western blot showing expression of PfPKAc-GFPDD in late schizonts cultured under protein- and RNA-depleting (−Shield-1/+GlcN) and control conditions (+Shield-1/−GlcN). Lysates derived from an equal number of parasites were loaded per lane. The membrane was first probed with α-GFP followed by α-GAPDH control antibodies. MW PfPKAc-GFP = 67.3 kDa, MW PfGAPDH = 36.6 kDa. Dashed lines mark the blot sections shown in Fig 1A. **(C)** Expression of PfPKAc-GFPDD in mid and late schizonts under protein- and RNA-stabilising conditions (+Shield-1/−GlcN) as assessed by live cell fluorescence imaging. Parasites were previously synchronised to an 8-hour window and imaged at 32 to 40 hpi and 40 to 48 hpi. Representative fluorescence images are shown. Parasite DNA was stained with Hoechst. Scale bar = 5 μm. DIC, differential interference contrast; hpi, hours postinvasion; WT, wild-type.
(TIF)

**S2 Fig. Gating strategy of flow cytometry data obtained from parasite multiplication assays.** Representative flow cytometry plots of a parasite culture (top frame; NF54/AP2-G-mScarlet/PKAc cKD, +Shield-1/−GlcN) and an uninfected RBC control sample (bottom frame) on day 1 of the multiplication assay are shown. Events were consecutively gated for the expected cell size, singlets and infected (SYBR green positive) RBCs. The resulting parasite multiplication plots are shown in Figs 1, 2, and 4–6 and in S6, S9, and S11 Figs. RBC, red blood cell.
(TIF)

**S3 Fig. SCRs of NF54/AP2-G-mScarlet/PKAc cKD and NF54/AP2-G-mScarlet control parasites. (A)** Left panel: SCRs of NF54/AP2-G-mScarlet/PKAc cKD parasites cultured under protein- and RNA-depleting (−Shield-1/+GlcN) (turquoise) and control conditions (+Shield-1/−GlcN) (blue). Right panel: SCRs of NF54/AP2-G-mScarlet control parasites cultured in the presence (+GlcN) (light grey) or absence of GlcN (−GlcN) (dark grey). SCRs were determined by high content imaging and automated image analysis by assessing PfAP2-G-mScarlet positivity among the total number of Hoechst-stained iRBCs. For each experiment, at least 607 Hoechst-positive cells were assessed for PfAP2-G-mScarlet expression. Open squares represent data points for individual replicates and the means and SD (error bars) of 3 biological replicate experiments are shown. Differences in SCRs have been compared using a paired 2-tailed Student $t$ test (statistical significance cutoff: $p < 0.05$). The raw data are available in the source data file (S2 Data). **(B)** Mean fold change in SCRs of NF54/AP2-G-mScarlet/PKAc cKD parasites cultured under −Shield-1/+GlcN compared to +Shield-1/−GlcN conditions (dark blue) and of NF54/AP2-G-mScarlet control parasites cultured under +GlcN compared to −GlcN conditions (black). Differences in the fold change in SCRs between NF54/AP2-G-mScarlet/PKAc cKD and NF54/AP2-G-mScarlet control parasites have been compared using an unpaired 2-tailed Student $t$ test (statistical significance cutoff: $p < 0.05$). S, Shield-1; GlcN, glucosamine. The raw data are available in the source data file (S2 Data). iRBC, infected red blood cell; SCR, sexual commitment rate.
(TIF)

**S4 Fig. Gametocyte maturation, exflagellation, and retention rates of NF54/AP2-G-mScarlet/PKAc cKD parasites. (A)** Representative images captured from Giemsa-stained blood smears showing the distinct morphology of stage I to V gametocytes cultured under PfPKAc-

GFPDD-depleting (−Shield-1/+GlcN) and control conditions (+Shield-1/−GlcN) over 11 days of maturation. Synchronous parasites were split (±Shield-1/±GlcN) as sexual/asexual ring stage parasites 24 hours after the induction of sexual commitment in the preceding IDC. To eliminate asexual parasites, gametocytes were cultured in +SerM supplemented with 50 mM GlcNAc from day 1 to 6 of gametocytogenesis. Scale bar = 5 μm. dgd, day of gametocyte development. **(B)** Full size western blot showing expression of PfPKAc-GFPDD in mature stage V gametocytes (day 11) under protein- and RNA-depleting (−Shield-1/+GlcN) and control conditions (+Shield-1/−GlcN). Lysates derived from an equal number of parasites were loaded per lane. The membrane was first probed with α-GFP followed by α-GAPDH control antibodies. MW PfPKAc-GFPDD = 79.8 kDa, MW PfGAPDH = 36.6 kDa. Dashed lines mark the blot sections shown in Fig 1D. **(C)** Full size western blot comparing expression of PfPKAc-GFPDD under protein- and RNA-depleting (−Shield-1/+GlcN), protein-depleting (−Shield-1/−GlcN) and control conditions (+Shield-1/−GlcN) in mature stage V gametocytes (day 11). Lysates derived from an equal number of parasites were loaded per lane. **(D)** Relative ExRs of NF54/AP2-G-mScarlet/PKAc cKD mature stage V gametocytes (day 14) cultured under protein- and RNA-depleting (−Shield-1/+GlcN) (purple) and control conditions (+Shield-1/−GlcN) (green) or under protein-depleting only (−Shield-1) (orange) and control conditions (+Shield-1) (blue). Open squares represent data points for individual replicates and the means and SD (error bars) of at least 3 biological replicate experiments are shown. Differences in ExRs have been compared using an unpaired 2-tailed Student $t$ test (statistical significance cutoff: $p < 0.05$). The raw data are available in the source data file (S2 Data). **(E)** Relative ExRs of mature NF54 WT stage V gametocytes (day 14) cultured in presence (+GlcN) (light grey) and absence of GlcN (−GlcN) (black). Parasites were cultured and the results obtained from 3 biological replicate experiments analysed as described in panel D. The raw data are available in the source data file (S2 Data). **(F)** Retention rates of NF54 WT stage III (day 6) and stage V (day 11) gametocytes cultured in absence (−Shield-1) (orange) and presence of Shield-1 (+-Shield-1) (blue). Coloured squares represent data points for individual replicates and the means and SD (error bars) of one biological replicate experiment with 6 technical replicates each are shown. Differences in retention rates have been compared using an unpaired 2-tailed Student $t$ test (statistical significance cutoff: $p < 0.05$). The raw data are available in the source data file (S2 Data). ExR, exflagellation rate; GlcN, glucosamine; IDC, intraerythrocytic developmental cycle; S, Shield-1; WT, wild-type.
(TIF)

**S5 Fig. CRISPR/Cas-9–based engineering of the NF54/PKAc cOE line.** Top: Scheme depicting the WT *glp3* target locus, the donor (pD_pkac_cOE) and pBF_gC-cg6 suicide constructs transfected into NF54 WT parasites to generate the NF54/PKAc cOE parasite line and the edited *glp3* locus. Primers used for diagnostic PCRs are indicated. Bottom: Results of PCR reactions performed on gDNA of 2 clones (M1 and M2) of the NF54/PKAc cOE line and NF54 WT control parasites confirm successful insertion of the PfPKAc cOE cassettes into the *glp3* locus. cOE, conditional overexpression; WT, wild-type.
(TIF)

**S6 Fig. Overexpression of PfPKAc-GFP and multiplication rates of NF54/PKAc cOE M1, M2, and S1 parasites. (A)** Expression of PfPKAc-GFP in NF54/PKAc cOE M2 parasites under overexpression-inducing (−GlcN) and control conditions (+GlcN) as assessed by live cell fluorescence imaging and western blot analysis. Synchronous parasites (0 to 8 hpi) were split (±GlcN) 40 hours before sample collection. Representative fluorescent images are shown. Parasite DNA was stained with Hoechst. Scale bar = 5 μm. For western blot analysis, parasite lysates derived from equal numbers of parasites were loaded per lane. The membrane was first

probed with α-GFP followed by α-GAPDH control antibodies. MW PfPKAc-GFP = 67.3 kDa, MW PfGAPDH = 36.6 kDa. The full size western blot is shown. **(B)** Full size western blot showing expression of PfPKAc-GFP in NF54/PKAc cOE M1 parasites under overexpression-inducing (–GlcN) and control conditions (+GlcN) conditions. Parasites were cultured and samples prepared as described in panel A. The membrane was first probed with α-GFP followed by α-GAPDH control antibodies. MW PfPKAc-GFPDD = 79.8 kDa, MW PfGAPDH = 36.6 kDa. Dashed lines mark the blot sections shown in Fig 2A. **(C)** Full size western blot shows expression of PfPKAc-GFP in NF54/PKAc cOE S1 parasites under overexpression-inducing (–GlcN) and control (+GlcN) conditions. Parasites were cultured and samples prepared as described in panel A. The membrane was first probed with α-GFP followed by α-GAPDH control antibodies. MW PfPKAc-GFPDD = 79.8 kDa, MW PfGAPDH = 36.6 kDa. Dashed lines mark the blot sections shown in Fig 2E. MW PfPKAc-GFP = 67.3 kDa, MW PfGAPDH = 36.6 kDa. **(D)** Parasite multiplication rates of NF54/PKAc cOE M1 (left) and S1 survivor parasites (right) under overexpression-inducing (–GlcN) and control conditions (+GlcN) over 2 generations. Open squares represent data points for individual replicates and the means and SD (error bars) of 3 biological replicates are shown. Differences in multiplication rates have been compared using a paired 2-tailed Student $t$ test (statistical significance cutoff: $p < 0.05$). Note that the same data is presented as an increase in parasitaemia over time in Fig 2D and 2G. The raw data are available in the source data file (S2 Data). cOE, conditional overexpression; DIC, differential interference contrast; hpi, hours postinvasion. (TIF)

**S7 Fig. Sequence alignment of PDK1 orthologs.** Clustal Omega [96] multiple sequence alignment of PfPDK1 (Pf3D7_1121900/UniProt ID Q8IIE7) and PvPDK1 (PVX_091715/UniProt ID A5K4N1) with well-characterised PDK1 homologues from *Arabidopsis thaliana* (UniProt ID Q9XF67), *Caenorhabditis elegans* (UniProt ID Q9Y1J3), and humans (UniProt ID O15530). Residues are highlighted in blue gradient depending on fractional conservation. The kinase catalytic domain spans residues 82 to 342 in human PDK1 [52] (purple section underneath the sequences), which also includes a carboxyl-terminal PH domain spanning residues 446–548 [51] (green section). Arrowheads denote residues forming the PIF-binding pocket in human PDK1 [22], asterisks denote residues mutated in PfPKAc OE survivors identified in this study and red bars denote residues that form part of the ATP-binding cleft. PH, pleckstrin homology. (TIF)

**S8 Fig. CRISPR/Cas-9–based engineering of the NF54/PDK1 cKD parasite line.** Top: Scheme depicting the WT *pfpdk1* locus, the donor (pD_pdk1-gfpdd) and the suicide (pHF_gC_pdk1-gfpdd) constructs transfected into NF54 WT parasites to generate the NF54/ PDK1 cKD parasite line and the edited *pfpdk1* locus. Primers used for diagnostic PCRs are indicated. Middle: Results of PCR reactions performed on gDNA of NF54/PDK1 cKD and NF54 WT control parasites confirm correct editing of the *pfpdk1* locus as well as plasmid concatemer integration. Bottom: Schematic map illustrating the integration of a donor plasmid concatemer based on double-crossover recombination of nonadjacent HRs on the concatemer. To simplify the schematic, the integration of a tandem assembly only is shown. HR, homology region; WT, wild-type. (TIF)

**S9 Fig. Expression of PfPDK1-GFPDD in NF54/PDK1 cKD parasites and multiplication rates of NF54 WT parasites in presence and absence of Shield-1. (A)** Expression of PfPDK1-GFPDD in ring (18 to 24 hpi), trophozoite (24 to 30 hpi), and schizont (42 to 48 hpi)

stages of NF54/AP2-G-mScarlet/PDK1 cKD parasites under protein-stabilising (+Shield-1) conditions as assessed by western blot analysis. Lysates derived from equal numbers of parasites were loaded per lane. The membrane was first probed with α-GFP followed by α-GAPDH control antibodies. MW PfPDK1-GFPDD = 101.1 kDa, MW PfGAPDH = 36.6 kDa. The full size western blot is shown. **(B)** Full size western blot shows expression of PfPDK1-GFPDD in NF54/AP2-G-mScarlet/PDK1 cKD parasites under protein-depleting (−Shield-1) and control (+Shield-1) conditions. Synchronous parasites (0 to 8 hpi) were split (±Shield-1) 40 hours before collection of the samples. Lysates derived from equal numbers of parasites were loaded per lane. The membrane was first probed with α-GFP followed by α-GAPDH control antibodies. MW PfPDK1-GFPDD = 101.1 kDa, MW PfGAPDH = 36.6 kDa. Dashed lines mark the blot sections shown in Fig 4B. **(C)** Increase in parasitaemia (left) and corresponding parasite multiplication rates (right) of NF54 WT parasites cultured in presence (+Shield-1) and absence of Shield-1 (−Shield-1). Synchronous parasites (0 to 6 hpi) were split (±Shield-1) 18 hours before the first measurement in generation 1. Open squares represent data points for individual replicates and the means and SD (error bars) of 3 biological replicates are shown. Differences in multiplication rates have been compared using a paired 2-tailed Student $t$ test (statistical significance cutoff: $p < 0.05$). The raw data are available in the source data file (S2 Data). hpi, hours postinvasion; WT, wild-type. (TIF)

**S10 Fig. SCRs, gametocytogenesis, and male gametogenesis of NF54/PDK1 cKD parasites.** **(A)** SCRs of NF54/PDK1 cKD parasites cultured in the presence (+Shield-1) or absence of Shield-1 (−Shield-1). Open squares represent data points for individual replicates and the means and SD (error bars) of 3 biological replicate experiments are shown. Differences in SCRs have been compared using a paired 2-tailed Student $t$ test (statistical significance cutoff: $p < 0.05$). The raw data are available in the source data file (S2 Data). **(B)** Representative images captured from Giemsa-stained thin blood smears showing the distinct morphology of stage I to V gametocytes cultured under PfPDK1-GFPDD-depleting (−Shield-1) and control (+Shield-1) conditions over 11 days of maturation. Synchronous parasites were split (±Shield-1) as sexual/asexual ring stage parasites 24 hours after the induction of sexual commitment in the preceding IDC. To eliminate asexual parasites, gametocytes were cultured in +SerM supplemented with 50 mM GlcNAc from day 1 to 6 of gametocytogenesis. Scale bar = 5 μm. dgd, day of gametocyte development. **(C)** Relative ExRs of mature NF54/PDK1 cKD stage V gametocytes (day 14) cultured in presence (+Shield-1) and absence of Shield-1 (−Shield-1). Synchronous parasites were split (±Shield-1) and cultured as described in panel B. Open squares represent data points for individual replicates and the means and SD (error bars) of 3 biological replicate experiments are shown. Differences in ExRs have been compared using an unpaired 2-tailed Student $t$ test (statistical significance cutoff: $p < 0.05$). The raw data are available in the source data file (S2 Data). ExR, exflagellation rate; IDC, intraerythrocytic developmental cycle; SCR, sexual commitment rate. (TIF)

**S11 Fig. CRISPR/Cas-9–based engineering and characterisation of the NF54/PKAc cOE M1/PDK1_mut and NF54/PKAc cOE S1/PDK1_wt lines. (A)** Top: Scheme depicting the *pfpdk1* locus of NF54/PKAc cOE S1 or M1 parasites, the donor (pD_S1rev or pD_M1mut) and the suicide (pHF_gC_S1rev or pHF_gC_M1mut) constructs transfected into either NF54/PKAc cOE S1 or NF54/PKAc cOE M1 parasites to generate the NF54/PKAc cOE S1/PDK1_wt and NF54/PKAc cOE M1/PDK1_mut parasite line, respectively, and the edited *pfpdk1* locus. Bottom: Sanger sequencing results of modified *pfpdk1* genes after targeted mutagenesis in NF54/PKAc cOE S1/PDK1_wt and NF54/PKAc cOE M1/PDK1_mut parasites confirms

correct editing. The expected sequences after successful editing, the corresponding amino acid changes and sequencing chromatograms are indicated. The asterisk marks the mutated residues (M51R or R51M). Capital letters highlight the synonymous nucleotide substitutions introduced by CRISPR/Cas-9 editing to destroy the sgRNA target site and to introduce the aspired amino acid change. **(B)** Full size western blots showing expression of PfPKAc-GFP in NF54/PKAc cOE S1/PDK1_wt (left) and NF54/PKAc cOE M1/PDK1_mut (right) parasites under OE-inducing (–GlcN) and control conditions (+GlcN). Synchronous parasites (0 to 8 hpi) were split (±GlcN) 40 hours before sample collection. Lysates derived from equal numbers of parasites were loaded per lane. The membranes were first probed with α-GFP followed by α-GAPDH control antibodies. MW PfPKAc-GFP = 67.3 kDa, MW PfGAPDH = 36.6 kDa. Dashed lines mark the blot sections shown in Fig 5A and 5B. **(C)** Parasite multiplication rates of NF54/PKAc cOE S1/PDK1_wt (top) and NF54/PKAc cOE M1/PDK1_mut (bottom) parasites under OE-inducing (–GlcN) and control conditions (+GlcN) over 2 generations. Open squares represent data points for individual replicates and the means and SD (error bars) of 3 biological replicates are shown. Differences in multiplication rates have been compared using a paired 2-tailed Student $t$ test (statistical significance cutoff: $p < 0.05$). Note that the same data is presented as an increase in parasitaemia over time in Fig 5C and 5D. The raw data are available in the source data file (S2 Data). cOE, conditional overexpression; hpi, hours postinvasion; sgRNA, single guide RNA; WT, wild-type.
(TIF)

**S12 Fig. Integration of multiple PfPKAc cOE cassettes into the *glp3* locus in NF54/PKAc cOE M1, M2, and S1-S6 parasites.** Top: Scheme depicting the WT *glp3* target locus and the pD_pkac_cOE donor plasmid used to integrate a PfPKAc cOE cassette into the *glp3* locus in NF54 WT parasites to generate the NF54/PKAc cOE parasite line (see also S5 Fig). Bottom: The boxed schematic illustrates the integration of pD_pkac_cOE donor plasmid concatemers into the *glp3* locus based on double-crossover recombination of nonadjacent HRs on the concatemer. For reasons of simplicity, the integration of a tandem assembly only is shown. *n*-1, number of integrated donor plasmids. Estimated mean copy numbers of integrated PfPKAc cOE cassettes (*n*) are shown for the 2 unselected NF54/PKAc cOE clones (M1, M2) and the 6 independently grown survivor populations (S1-S6), alongside the PfPDK1 mutations identified in the 6 NF54/PKAc cOE survivors (see also Fig 3). Copy numbers of PfPKAc cOE cassettes were calculated from WGS data and the analysis steps are described in the Materials and Methods section. cOE, conditional overexpression; HR, homology region; WGS, whole genome sequencing; WT, wild-type.
(TIF)

**S13 Fig. CRISPR/Cas-9–based engineering and characterisation of the NF54/PKAcT189V cOE line.** **(A)** Top: Scheme depicting the WT *glp3* target locus, the donor (pD_pkacT189V_cOE) and pBF_gC-cg6 suicide constructs transfected into NF54 WT parasites to generate the NF54/PKAcT189V cOE parasite line and the edited *glp3* locus. Primers used for diagnostic PCRs are indicated. Bottom: Results of PCR reactions performed on gDNA of the NF54/PKAcT189V cOE line and NF54 WT control parasites confirm successful insertion of the PfPKAcT189V cOE cassette into the *glp3* locus. **(B)** Full size western blot showing expression of PfPKAcT189V-GFP in NF54/PKAcT189V cOE parasites under OE-inducing (–GlcN) and control conditions (+GlcN). Synchronous parasites (0 to 8 hpi) were split (±GlcN) 40 hours before collection of the samples. Lysates derived from equal numbers of parasites were loaded per lane. The membrane was first probed with α-GFP followed by α-GAPDH control antibodies. MW PfPKAcT189V-GFP = 67.3 kDa, MW PfGAPDH = 36.6 kDa. Dashed lines mark the blot sections shown in Fig 6A. cOE, conditional overexpression; hpi, hours postinvasion; WT,

wild-type.
(TIF)

**S14 Fig. Activity of inhibitors of human PDK1 on *P. falciparum* blood stage parasite multiplication. (A, B)** $IC_{50}$ values for the 3 human PDK1 inhibitors BX-795, BX-912, and GSK2334470 and the 2 antimalarial control compounds Chloroquine and Artesunate on the multiplication of NF54 WT parasites (A) and NF54/PDK1 cKD and NF54 WT parasites cultured in the absence (−Shield-1) or presence of Shield-1 (+Shield-1) (B). Symbols represent individual $IC_{50}$ values calculated from 2 technical replicate dose response assays each. Drug dose response assays were performed in 3 biological replicates, means and SD (error bars) are indicated. The raw data are available in the source data file (S2 Data). **(C)** Representative images captured from Giemsa-stained thin blood smears of NF54 WT parasites exposed to BX-795, BX-912, and GSK2334470 (10x $IC_{50}$) and the solvent control (DMSO). Drugs were added to synchronous young ring stage parasites (0 to 4 hpi) and blood smears prepared every 12 hours for 60 hours. The curved arrow marks the time point of merozoite release and invasion into new RBCs in the DMSO control population. **(D)** Representative images captured from Giemsa-stained thin blood smears of NF54 WT parasites exposed to BX-795, BX-912, and GSK2334470 (10x $IC_{50}$) and the DMSO solvent control. Drugs were added to synchronous early ring stages (0 to 4 hpi) (left panel), late ring stages/early trophozoites (20 to 24 hpi) (middle panel), or late trophozoites/early schizonts (30 to 34 hpi) (right panel) for 18 hours, followed by drug washout and further culturing in drug-free medium. Blood smears were prepared 18, 30 and, 42 hours after the start of the assay. The curved arrow marks the time point of merozoite release and invasion into new RBCs in the DMSO control population. ER, early ring; ES, early schizont; hpi, hours postinvasion; LR/LR_g2, late rings/late rings generation 2; LR, late ring; LS, late schizont; MR/MR_g2, mid rings/mid rings generation 2; MS, mid schizont; RBC, red blood cell; T, trophozoites; T/T_g2, trophozoites/trophozoites generation 2; WT, wild-type.
(TIF)

**S1 Table. Oligonucleotides used for cloning of transfection constructs.** Names and sequences of oligonucleotides, plasmids and cell lines are indicated. Sequences essential for Gibson assembly reactions (Gibson overhangs) or for T4 DNA ligase-dependent cloning of double-stranded sgRNA-encoding fragments (5′ and 3′ overhangs) are highlighted with capital letters. Italicised letters highlight the annealed sequences (sgRNAs) and colour-highlighted letters represent introduced sequence mutations. sgRNA, single guide RNA.
(PDF)

**S2 Table. Primers used for diagnostic PCRs on gDNA of transgenic parasite lines.** Names and sequences of oligonucleotides and cell lines are indicated.
(PDF)

**S1 Data. Multiple nucleotide sequence alignment of Pf3D7_1121900/*pfpdk1* sequences of the NF54/PKAc cOE clones M1 and M2 and the 6 PfPKAc OE-tolerant survivor populations S1-S6 determined by WGS.** The *pfpdk1* coding sequences of the NF54/PKAc cOE M1 and M2 clones are identical to the Pf3D7_1121900 reference sequence retrieved from PlasmoDB (www.plasmodb.org) (top row). *pfpdk1* coding sequences of the NF54/PKAc cOE survivor populations S1-S6 are shown and deviations from the reference sequence are highlighted in green. NF54/PKAc cOE survivor S6 consists of 2 subpopulations with one carrying the c.252A>T mutation and the other one carrying the c.491A>G mutation (as verified by inspection of the sequencing read pairs). cOE, conditional overexpression; WGS, whole genome

sequencing.
(PDF)

**S2 Data. Source data for all the graphs and charts presented in the main and Supporting information figures.** This file contains separate worksheets. Each worksheet lists the raw data underlying the graphs or charts presented in the main figure panels (Figs 1B, 1E, 2C, 2D, 2F, 2G, 4C, 5C, 5D, and 6B) and Supporting information figures (S3, S4, S6, S9, S10, S11, and S14 Figs).
(XLSX)

## Acknowledgments

We thank Christian Flueck for his valuable inputs on the manuscript and Judith Straimer for helpful advice on the drug assays.

## Author Contributions

**Conceptualization:** Eva Hitz, Till S. Voss.

**Data curation:** Eva Hitz, Natalie Wiedemar, Till S. Voss.

**Formal analysis:** Eva Hitz, Natalie Wiedemar, Armin Passecker, Ioannis Vakonakis, Till S. Voss.

**Funding acquisition:** Eva Hitz, Ioannis Vakonakis, Pascal Mäser, Till S. Voss.

**Investigation:** Eva Hitz, Natalie Wiedemar, Armin Passecker, Beatriz A. S. Graça, Christian Scheurer, Ioannis Vakonakis.

**Methodology:** Eva Hitz, Natalie Wiedemar, Ioannis Vakonakis, Till S. Voss.

**Project administration:** Eva Hitz, Sergio Wittlin, Nicolas M. B. Brancucci, Ioannis Vakonakis, Pascal Mäser, Till S. Voss.

**Resources:** Sergio Wittlin, Nicolas M. B. Brancucci, Ioannis Vakonakis, Pascal Mäser, Till S. Voss.

**Supervision:** Sergio Wittlin, Nicolas M. B. Brancucci, Pascal Mäser, Till S. Voss.

**Validation:** Eva Hitz, Natalie Wiedemar, Armin Passecker, Beatriz A. S. Graça, Christian Scheurer, Till S. Voss.

**Visualization:** Eva Hitz, Natalie Wiedemar, Beatriz A. S. Graça, Ioannis Vakonakis, Till S. Voss.

**Writing – original draft:** Eva Hitz, Till S. Voss.

**Writing – review & editing:** Eva Hitz, Natalie Wiedemar, Armin Passecker, Beatriz A. S. Graça, Christian Scheurer, Sergio Wittlin, Nicolas M. B. Brancucci, Ioannis Vakonakis, Pascal Mäser, Till S. Voss.

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
