## [Editor Report · Decision Letter 0]

2 Jul 2021

Dear Dr. Voss, 

Thank you for submitting your manuscript entitled "The 3-phosphoinositide-dependent protein kinase 1 is an essential upstream activator of protein kinase A in malaria parasites" for consideration as a Research Article by PLOS Biology.

Your manuscript has now been evaluated by the PLOS Biology editorial staff, as well as by an academic editor with relevant expertise, and I am writing to let you know that we would like to send your submission out for external peer review. In order to move forward with the manuscript after review, we would need clear support from the reviewers that the work reports the type of conceptual advance that would justify publication in PLOS Biology.

Please re-submit your manuscript within two working days, i.e. by Jul 04 2021 11:59PM.

Kind regards,

Paula

---

Paula Jauregui, PhD

Associate Editor

PLOS Biology

---

## [Decision Letter · Decision Letter 1]

20 Aug 2021

Dear Dr. Voss,

Thank you very much for submitting your manuscript "The 3-phosphoinositide-dependent protein kinase 1 is an essential upstream activator of protein kinase A in malaria parasites" for consideration as a Research Article at PLOS Biology. Your manuscript has been evaluated by the PLOS Biology editors, an Academic Editor with relevant expertise, and by several independent reviewers.

In light of the reviews (below), we are pleased to offer you the opportunity to address the comments from the reviewers in a revised version that we anticipate should not take you very long. We will then assess your revised manuscript and your response to the reviewers' comments and we may consult the reviewers again.

In particular, we are overruling reviewer #1's first set of concerns on the rigidity issue, but we think it is important that you address reviewer #1's concern on whether the two proteins truly interact in vivo. Reviewer #1 suggests several experiments, one of which you could do to demonstrate direct interaction, or by discussing the caveat for your conclusion as pointed out by the reviewer. Please also address the comments the reviewers make on the writing and clarifications on the text. 

We expect to receive your revised manuscript within 1 month.

**IMPORTANT - SUBMITTING YOUR REVISION**

*Resubmission Checklist*

*Published Peer Review*

*PLOS Data Policy*

*Blot and Gel Data Policy*

Sincerely,

Paula

---

Paula Jauregui, PhD

Associate Editor

PLOS Biology

REVIEWS:

Reviewer #1: Signal transduction pathways in Plasmodium.

Reviewer #2: Wai-Hong Tham. Host-parasite interactions.

Reviewer #1: This is a really interesting manuscript investigating the role of the catalytic subunit of the cAMP-dependent protein kinase (PKAc) in Plasmodium falciparum asexual blood stage development and sexual development as well as its regulation. The study uses a number of approaches including conditional overexpression and conditional knockdown to confirm the role of PKAc in erythrocyte invasion, but also to show no apparent essential role in gametocyte commitment or development. The central hypothesis of the study is that PKAc is activated by an orthologue of the 3-phosphoinositide-dependent protein kinase 1 (PDK1). Overall the manuscript represents a huge amount of work involving many excellent experiments carried out very carefully and that are very well presented. The manuscript is also very nicely written and includes helpful summary statements in the individual results sections keeping the reader on board throughout. My additional thoughts and concerns are detailed below.

Figure 1. As the authors point out, there is previous literature implicating cAMP/PKA in gametocyte deformability and so I think that these experiments were a good idea. However, the differences in gametocyte retention using microsphiltration +/- Shield 1 shown in the plot, seem very small. Visually they appear non-significant, although it is stated that the differences are significant using an unpaired two-tailed Student's t test (p = 0.04). The SD values for stage V gametocytes seem quite large. It would be nice to see how big the differences in retention are using a 'positive control' (with WT gametocytes) that has been shown previously to change gametocyte retention significantly. This would help to determine whether the observed level of change is comparable with what has been found previously. I note that only one biological replicate (with 6 technical replicates) of the control experiment with WT gametoctyes +/- Shield 1 has been presented which doesn't increase confidence in the this part of the work. As it stands it is hard to judge the biological relevance of these observations with such a small difference in retention upon depletion of such a key signalling molecule. I think without additional experiments to increase confidence in this part of the work, the authors should consider removing the data from the manuscript.

Overexpression of an ectopic PKAc driven by a constitutive promoter was lethal and gave rise to parasites which stalled at the ~2 nuclei early schizont stage. Importantly the timing of this overexpression phenotype is very different to that reported previously using conditional knockout that showed a role for PKAc in erythrocyte invasion. However, conditional knockdown of PKAc confirmed its role in invasion. Interestingly, in the PKAc overexpression experiment, parasites reappeared in cultures again after 2 weeks; they still over-expressed PKAc and there was no mutation in either the ectopic or endogenous PKAc. These parasites showed normal nuclei numbers and blood stage development.

It is really intriguing that in the parasites selected, that could tolerate overexpression of PKAc, WGS showed mutations only in the putative PDK1 gene. (At the top of page 9, why are the M1 and M2 clones referred to as 'unselected clones'? I think that the term is confusing and should be deleted and the clones just referred to as M1 and M2 that have been described earlier in the text).

The homology model of the PfPDK1 orthologue predicts nicely where the mutant residues are situated and how they are thought to affect catalytic activity, but not interaction with other proteins. 

No PDK1 null could be obtained, but the conditional knockdown was non-essential suggesting that residual protein was sufficient for parasite development.

Does the PDK1 knockout/knockdown work add anything to the paper, especially because the results of the two approaches taken are conflicting? The inability to generate a null mutant suggests that PDK1 is essential in blood stages consistent with cited published work whereas the knockdown shows no phenotype, leading the authors to conclude that residual protein allows the parasite to develop normally. However, the level of knockdown (Figure 4b) looks very good to me. Could the findings of these experiments be just mentioned in the Discussion rather than presenting the results? Do the authors have any thoughts on why the multiplication rate is higher in the knockdown parasites?

The main issue I have with the manuscript is as follows. On the one hand if PDK1 is indeed a physiological regulator of PKAc in P. falciparum, this has been a fantastic approach leading to its elucidation that will surely be a good approach for other researchers to follow in future to identify regulatory proteins. On the other hand unfortunately, it seems plausible that selection of these parasites with PDK1 mutations has taken place simply because they can survive the lethal effects of this artificial PKAc overexpression that does not occur at this life cycle stage in WT parasites and that the PDK1 orthologue is not actually a regulator of PKAc. 

Therefore I don't agree with the statement (on page 12 under the subheading) : 'To confirm the specific function of PfPDK1 in regulating PfPKAc activity…etc'. This is because in this section the authors further demonstrate the importance of the PDK1 M51R mutation in mediating parasite survival of the PfPKAc-GFP overexpression line by generating additional transgenic lines. In the summary of this section of the results on page 13, there is a more cautious statement that I do agree with: '…suggest that PfPDK1 is the kinase that phosphorylates and activates PfPKAc'.

Even though the presence (and amplification) of PDK1 mutations in the survivor parasites is compelling, what is lacking in the manuscript is an experiment that shows activation of PKAc by PDK1 or their mutual interaction in wild type parasites. This would be needed to provide direct evidence that PDK1 is a physiological regulator of PKAc and to dismiss the nagging suspicion/possibility that this may be an artificial survival mechanism in parasites selected due to overexpression of PKAc at the wrong* life cycle stage (*the essential role of PKA is in invasion, the lethal phenotype here is in early schizonts). Whilst I agree that the results strongly imply that PKAc and PDR1 should be able to interact in WT parasites, the question is: do they actually ever interact with each other in WT parasites? 

Regarding demonstration of the interaction, I am really not a believer in the value of showing phosphorylation of recombinant proteins by a recombinant kinase as I think the results of this type of experiment can be misleading because in vitro some kinases will phosphorylate any protein with a consensus substrate site. This would give no information on whether the two proteins are normally expressed at the same time and in the same place. It would be great if a pull down or e.g. a BioID approach was able to demonstrate a direct interaction between PKAc and PDK1 in both WT and survivor mutant parasites to help provide this missing link. 

Another experiment that would increase confidence in the hypothesis that PDK1 is a physiological PKAc activator is to try to complement the PKAcDD (endogenous locus) knockdown line with an ectopic copy of PKAc (under the control of the PKAc promoter) that carries the T189V substitution (in parallel with a control experiment adding an ectopic WT copy). If the hypothesis is correct it should not be possible to achieve complementation with the T189V copy as PKD1 would not be able to activate this mutant PKAc and invasion would not be possible.

Even though the survivor population are fully viable, if mutant PDK1 is less able to phosphorylate/activate PKAc, you might expect a reduced erythrocyte invasion efficiency (since PKAc is essential for invasion) in the survivor parasites. Has the efficiency of invasion been measured in survivor parasites compared to WT to check this possibility? Any difference to WT would be interesting as it would be a link between mutant PDK1 and invasion.

I note that PDK1 is a relatively small kinase (525 amino acids). It would be nice to know in future how the mutations affect catalytic activity (as hypothesised) compared to WT. Perhaps kinase assays could easily be done on immunoprecipitated existing tagged WT and mutant PDK1 isoforms?

A very nice experiment is carried out showing that overexpression of a non-phosphorylatable Thr/Val substitution in the PKAc activation loop (which in other species is the target of PDK1) was not lethal. This provides further confirmation that PKAc and the PDK1 orthologue interact in the overexpression lines. I have to say that although this is not the wild type situation, manipulation of this specific threonine residue to reverse the lethal phenotype of PKAc overexpression does add an extra layer of confidence that the two proteins are likely to be physiological partners.

Finally the activity (EC50 ~1 µM) of three inhibitors of human PDK1 on blood stage P. falciparum multiplication is demonstrated. However it would be useful if the authors could determine/clarify the stage at which these inhibitors are killing blood stage parasites to confirm whether or not this is consistent with PDK1 being their primary target. Either way I agree that PDK1 is worthy of additional future attention as a potential antimalarial drug target. 

In the Discussion, the first statement should be modified to reflect the strong possibility that phosphorylation of proteins additional to AMA1 will underlie the essential role of PfPKA.

The statement: '..we discovered the upstream kinase that is required for PfPKAc activation' seems a little strong since no direct evidence of an interaction between PKAc and PDK1 has been demonstrated in WT parasites. The statement should be softened to reflect the fact there is a degree of uncertainty as I have argued above..

Similarly, the later statement (top of page 18) beginning: 'Together these striking results demonstrate…'. I think the demonstrate should be changed to strongly suggest, but I agree that the results 'provide compelling evidence that PfPDK1 is the kinase that targets this residue'.

Similarly, the later statement (top page 19): 'we identified PfPDK1 as the upstream kinase activating PfPKAc…'.

Similarly at the end of the Introduction (page 5) 'and that activation of PfPKAc is strictly dependent on PfPDK1-mediated phosphorylation'.

Also the at the end of the Abstract: 'and identify PfPDK1 as a crucial upstream regulator in this pathway'.

All of these statements should be softened.

One thing that is surprisingly missing from the manuscript is any mention/discussion of how cAMP levels might impact the PfPDK1 activation process. I presume that the authors are thinking that activation of the overexpressed PKAc can happen independently of cAMP. I think that if there is insufficient PKAr expressed at this stage to form complexes with the excess PKAc, this could indeed be the case. However, in WT parasites, cAMP is obviously needed to activate PKAc (by binding to PKAr and causing dissociation of the complex allowing PKAc activation). I think this issue should be briefly dealt with in the Discussion. It will be interesting in future to explore the dynamics of the process by which PKAc is activated in the presence of cAMP and PDK1 in WT parasites, if it indeed turns out to be an essential upstream PKAc activator as the experiments in this study have strongly suggested.

Reviewer #2: The manuscript by Hitz et al uses elegant malaria genetic manipulations to understand the role of PfPKA catalytic domain in the sexual stages of the malaria parasites. While the authors find that PfPKAc does not play a major role in sexual commitment/gametocyte maturation, it does contribute to gametocyte rigidity (though possibly not the major driver). Using an overexpression system of PfPKAc, the exciting result is the "survivor" parasites have only compensatory mutations in PfPDK1 using whole genome sequencing, suggesting that a regulation network between these two kinases. The authors follow up on this premise with further predicted targeted selected mutations in genetically modified parasites to show that T189 in the PfPKAc activation loop is crucial. As with PfPKA, the authors show that PfPDK1 is also not involved in gametocyte/sexual commitment.

While the authors did not uncover a new function for both these kinases in gametocyte function, this paper represents a substantial amount of work that is extremely thorough and well validated. It is biologically relevant as all the work was performed in parasites vs recombinant kinases. In fact, it is quite hard to pick any further experiments or repeats that I would have like to see or required further interpretation. The authors should also be commended on not over stating any result that they have, especially around the gametocyte rigidity. In addition, while somewhat aside to their main results, they describe an important caveat using GlnN in studying sexual commitment and male gametocyte exflagellation.

My only minor comment is for the authors to perhaps provide a rationale on why they made the M51R transgenic versus any of the other mutations?

---

## [Editor Report · Decision Letter 2]

5 Nov 2021

Dear Dr. Voss,

Thank you for submitting your revised Research Article entitled "The 3-phosphoinositide-dependent protein kinase 1 is an essential upstream activator of protein kinase A in malaria parasites" for publication in PLOS Biology. I have now discussed your revision with the Academic Editor. 

We will probably accept this manuscript for publication, provided you satisfactorily address the following data and other policy-related requests.

Please ensure that figure legends in your manuscript include information on where the underlying data can be found, and ensure your supplemental data file/s has a legend.

We require the original, uncropped and minimally adjusted images supporting all blot and gel results reported in an article's figures or Supporting Information files. We will require these files before a manuscript can be accepted so please prepare and upload them now. Please carefully read our guidelines for how to prepare and upload this data: https://journals.plos.org/plosbiology/s/figures#loc-blot-and-gel-reporting-requirements. We need this for the blots in figures S6A and S9A. 

We expect to receive your revised manuscript within two weeks. 

*Published Peer Review History*

*Early Version*

Sincerely,

Paula 

---

Associate Editor,

pjaureguionieva@plos.org,

PLOS Biology

---

## [Editor Report · Decision Letter 3]

12 Nov 2021

Dear Dr. Voss,

On behalf of my colleagues and the Academic Editor, Boris Striepen, I am pleased to say that we can in principle accept your Research Article "The 3-phosphoinositide-dependent protein kinase 1 is an essential upstream activator of protein kinase A in malaria parasites" for publication in PLOS Biology, provided you address any remaining formatting and reporting issues. These will be detailed in an email that will follow this letter and that you will usually receive within 2-3 business days, during which time no action is required from you. Please note that we will not be able to formally accept your manuscript and schedule it for publication until you have any requested changes.

PRESS

Sincerely, 

Paula

---

Paula Jauregui, PhD 

Associate Editor 

PLOS Biology
